# HUB: A method to model and extract the distribution of ice nucleation temperatures from drop-freezing experiments

Ingrid de Almeida Ribeiro[1], Konrad Meister[2,3,4], Valeria Molinero[1]

[1]Department of Chemistry, The University of Utah, 315 South 1400 East, Salt Lake City, Utah 84112-0850, USA
[2]Max Planck Institute for Polymer Research, 55128 Mainz, Germany
[3]Department of Chemistry and Biochemistry, Boise State University, Boise, Idaho 83725, USA
[4]Biomolecular Sciences Graduate Programs, Boise State University, Boise, ID 83725, USA

*Correspondence to*: Valeria Molinero (Valeria.Molinero@utah.edu)

**Abstract.** The heterogeneous nucleation of ice is an important atmospheric process facilitated by a wide range of aerosols. Drop-freezing experiments are key for the determination of the ice nucleation activity of biotic and abiotic ice nucleators (INs). The results of these experiments are reported as the fraction of frozen droplets $f_{ice}(T)$ as a function of decreasing temperature, and the corresponding cumulative freezing spectra $N_m(T)$ computed using Vali's methodology. The differential freezing spectrum $n_m(T)$ is an approximant to the underlying distribution of heterogeneous ice nucleation temperatures $P_u(T)$ that represents the characteristic freezing temperatures of all IN in the sample. However, $N_m(T)$ can be noisy, resulting in a differential form $n_m(T)$ that is challenging to interpret. Furthermore, there is no rigorous statistical analysis of how many droplets and dilutions are needed to obtain a well-converged $n_m(T)$ that represents the underlying distribution $P_u(T)$. Here, we present the HUB method and associated Python codes that model (HUB-forward code) and interpret (HUB-backward code) the results of drop-freezing experiments. HUB-forward predicts $f_{ice}(T)$ and $N_m(T)$ from a proposed distribution $P_u(T)$ of IN temperatures, allowing its users to test hypotheses regarding the role of subpopulations of nuclei in freezing spectra, and providing a guide for a more efficient collection of freezing data. HUB-backward uses a stochastic optimization method to compute $n_m(T)$ from either $N_m(T)$ or $f_{ice}(T)$. The differential spectrum computed with HUB-backward is an analytical function that can be used to reveal and characterize the underlying number of IN subpopulations of complex biological samples (e.g. ice nucleating bacteria, fungi, pollen), and quantify the dependence of these subpopulations on environmental variables. By delivering a way to compute the differential spectrum from drop freezing data, and vice-versa, the HUB-forward and HUB-backward codes provide a hub to connect experiments and interpretative physical quantities that can be analysed with kinetic models and nucleation theory.

## 1 Introduction

Ice nucleators (INs) of biological and abiotic origins present in aerosols are responsible for facilitating the heterogeneous freezing of atmospheric water droplets above the homogeneous nucleation temperature (Murray et al., 2012; Demott et al.,

2016; Demott et al., 2003). The potential of these aerosols as ice nuclei has significant implications for cloud properties and precipitation patterns (Gettelman et al., 2012; Mülmenstädt et al., 2015; Froyd et al., 2022). Freezing experiments are key sources of information to determine the range of temperatures over which INs promote ice nucleation. The most common method to characterize INs is through immersion freezing experiments, for which a wide range of assays and instruments have been developed. A comprehensive report of various drop freezing techniques can be found in (Miller et al., 2021). The

assays are typically performed by placing uniformly sized water droplets with a known IN concentration or area on a substrate or in a multiwall plate that is gradually cooled from a temperature above 0$^{\mathrm{o}}$C until all droplets are frozen(Kunert et al., 2018; Budke and Koop, 2015). Droplet freezing is detected visually or through measurement of the latent heat release (Stratmann et al., 2004; Budke and Koop, 2015; Kunert et al., 2018; Reicher et al., 2018), allowing the assignment of a heterogeneous nucleation temperature to each droplet. Drop-freezing experiments record the fraction of frozen droplets,

$f_{ice}(T)$, as a function of decreasing temperature; for soluble or dispersible INs $f_{ice}(T)$ curves are typically collected at various ten-fold dilutions of the IN sample.

Historically, there have been two interpretations of the dispersion of nucleation temperatures in heterogeneous freezing experiments. The first approach suggests that the stochastic nature of the nucleation process dominates the variability in freezing temperatures (Bigg, 1953; Carte, 1956), while the second approach assumes that the dispersion in

temperatures mostly arises from a distribution of nucleation sites (Fletcher, 1969), each with a deterministic, singular nucleation temperature (Levine, 1950; Vali and Stansbury, 1966). Variability in the temperature, volume, and amount of ice-nucleating particles per droplet can also contribute to the dispersion of freezing temperatures (Vali, 2019; Knopf et al., 2020). There is consensus now that both stochastic effects and sample heterogeneities contribute to the distribution of freezing temperatures, and both approaches are used for the modelling of drop-freezing experiments (Vali, 1971; Marcolli et

al., 2007; Niedermeier et al., 2011; Murray et al., 2011; Broadley et al., 2012; Wright and Petters, 2013; Herbert et al., 2014; Harrison et al., 2016; Alpert and Knopf, 2016; Vali, 2019; Fahy et al., 2022a). Stochastic modelling of the freezing curves is based on predicting the survival probability of liquid water containing IN as a function of supercooling, and requires a model for the temperature dependence of the nucleation rate of the IN components. These models have been solved numerically or evolved with Monte Carlo simulations to interpret or resolve the distribution of ice nucleation properties of minerals

(Marcolli et al., 2007; Murray et al., 2011; Broadley et al., 2012; Wright and Petters, 2013; Herbert et al., 2014; Harrison et al., 2016) and organics (Zobrist et al., 2007; Alpert and Knopf, 2016) and to perform parametric bootstrapping of experimental data (Wright and Petters, 2013; Harrison et al., 2016). The advantage of the stochastic modelling approach is that it enables a direct link to microscopic properties of the nuclei and can account for the cooling rate dependence of the $f_{ice}(T)$ data. These approaches require the use of analytical models for the freezing rates and their distribution in the sample.

The modelling of freezing experiments based on the singular approach is based on the framework proposed by Vali (Vali, 1971). He assumed that each particular IN has a characteristic ice nucleation temperature that is independent of the cooling history. This implies that the IN with the highest characteristic nucleation temperature in a droplet is responsible for its freezing. Given a total number of droplets $N_0$, the number of frozen droplets $N_F(T)$ at a temperature $T$ gives the range of

characteristic freezing temperatures that determines the ice nucleation activity and is used to produce the cumulative freezing
spectrum (Vali, 1971; Vali, 2014, 2019),

$$N_m(T) = \frac{1}{X}[\ln N_0 - ln\, N_L(T)] = -\frac{1}{X}\ln[1 - f_{ice}(T)\,], \tag{1a}$$

where $N_L(T) = N_0 - N_F(T)$ is the number of unfrozen droplets, $f_{ice}(T) = N_F(T)/N_0$ is the fraction of frozen droplets at temperature $T$, and $X$ is a normalization factor per unit volume of water, unit mass, or surface of the INs (Vali, 2019). For soluble INs, the normalization factor is commonly defined by the mass of the ice nucleating material $X = \rho\,(V_{drop}/d)$, where $\rho$ is the density of the initial solution, $V_{drop}$ is the droplet volume and $d$ is the dilution factor (Kunert et al., 2018). The
IN surface area per drop, $X = A_{drop}$, is sometimes used as normalization factor for insoluble INs (e.g., dust, crystals), resulting in a cumulative spectrum per area denoted as $N_s(T)$. However, it is challenging to measure the total IN surface area accurately (Knopf et al., 2020). We note that **Eq. (1a)** can be used even when the absolute concentrations or areas of the IN are unknown, provided that the user knows the relative concentration of the dilution series derived from a parent sample. The differential freezing spectrum $n_m(T)$ is obtained by differentiation of the cumulative spectrum, (Vali, 1971)

$$n_m(T) = \frac{dN_m(T)}{dT} = -\frac{1}{XN_L(T)}\frac{dN_L(T)}{dT}\,. \tag{1b}$$

The differential spectrum identifies the density of IN active at each temperature, and was identified by Vali as the central quantity that can be derived and interpreted from drop-freezing experiments (Vali, 1971; Vali, 2019).

The determination of the differential spectrum from the cumulative one by finite differentiation is subject to significant noise, requiring a careful selection of the temperature intervals and extensive sampling (Vali, 2019). As stochastic effects are not considered in the singular temperature formalism, the cumulative and differential spectra should –in principle-
depend on the cooling rate (Vali, 1994). The stochastic nature of ice nucleation, combined with the uncertainties associated with the experimental measurements (e.g., different droplet volumes, inhomogeneous samples, different detection efficiencies), can produce significant variations in the cumulative freezing spectra, that result in large uncertainties in $n_m(T)$. Parametric and nonparametric bootstrapping based on the singular approximation and Monte Carlo simulations have been used to estimate confidence intervals in freezing spectra measurements (Vali, 2019; Fahy et al., 2022a; Fahy et al., 2022b).

A central assumption of the singular freezing approximation is that the freezing of a droplet containing multiple INs is promoted by the IN with the highest nucleation temperature (Levine, 1950). The extreme-value sampling is apparent in the concentration dependence of $f_{ice}(T)$ in experiments (Marcolli et al., 2007; Budke and Koop, 2015; Kunert et al., 2018; Lukas et al., 2022). Using probability theory, Levine demonstrated that if the distribution of ice nucleation temperatures of the IN population follows an exponential distribution, then the sampling of droplet freezing temperatures corresponds to a
Gumbel distribution, and the median freezing temperature $T_{\text{MED}}$ of the droplets scales with the logarithm of the number (or total nucleating area) of IN per droplet (Levine, 1950). Sear more recently demonstrated that Levine's approach is a particular solution for a generalized extreme-value problem, and used modern extreme value statistics to derive the scaling of $T_{\text{MED}}$ with the number of IN sites per droplet for the three generalized extreme value distributions (GEV): Gumbel that

would arise from an underlying IN distributions with exponential tails, Frechet from those with power law tails, and Weibull
from those with an upper cutoff in the freezing temperature of the IN (Sear, 2013). However, there are limitations for the use
of the analytical approaches of Sear and Levine for the interpretation of actual drop freezing data. First, the extreme value
sampling results in one of the three GEV only in the limit of extremely large number of IN per droplet, while in experiments
the sampling is typically performed over dilutions down to a few IN per droplet. There is no analytical formulation for the
dependence of the extreme value distribution in the low to intermediate concentration regime. Second, the analytical theory
assumes that the sampling is complete (i.e. the number of droplets is extremely large), while experiments are typically
performed with tens to hundreds of droplets. Third, Sear notes that there is no general analytical theory to predict the GEV
from a mixture of populations of nuclei with different temperature dependences (Sear, 2013). In this study we overcome
these three limitations through a numerical implementation of extreme-value statistics for the modelling of drop-freezing
experiments.

A consequence of extreme-value sampling is that the differential spectrum $n_m(T)$ represents the underlying
distribution of ice nucleation temperatures of all INs in the sample, which we denote as $P_u(T)$, only when the sampling of IN
in the drop-freezing experiments is complete. The underlying distribution $P_u(T)$ is akin to a hub that connects the
experimental freezing temperatures to physical analysis based on nucleation theory or kinetic and equilibrium models that
can elucidate the mechanisms and origins of the distributions of INs (**Fig. 1**). We here call the cumulative spectrum $N_m(T)$
obtained through **Eq. (1a)** in this complete sampling limit the intrinsic cumulative spectrum of the system, $I_u(T)$ (**Fig. 1**).
While there is consensus that the quality of the freezing spectrum increases with the number of droplets, a rigorous analysis
of how many droplets and IN dilutions should be measured to provide accurate freezing spectra is still lacking. The first goal
of the present study is to provide a strategy to optimize the sampling of drop-freezing experiments to derive interpretable
differential spectra that is a good approximant of the underlying distribution of heterogeneous ice nucleation temperatures of
the sample.

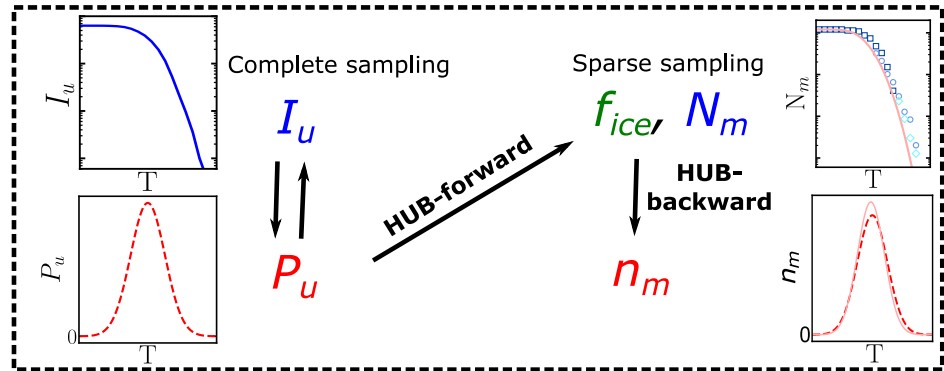

**Figure 1: Diagram illustrating the usage of the HUB code: $n_m(T)$ are obtained from the sparsely sampled $f_{ice}(T)$ or $N_m(T)$
through the HUB-backward, the effect on $f_{ice}(T)$ or $N_m(T)$ obtained from the complete sampling of the underlying distribution**
$P_u(T)$ **through HUB-forward. The intrinsic cumulative spectrum $I_u(T)$ is proportional to $\int_{T_m}^{T} P_u(T')dT'$ (Section 2.2).**

The existence of subpopulations or classes in the population of INs (e.g. different classes of bacterial INs, different ice nucleating sites on complex materials like dust) (Turner et al., 1990) is common in atmospheric aerosols. While several studies have broadly defined populations from the cumulative spectra by the range of nucleation temperatures they encompass (Turner et al., 1990; Creamean et al., 2019) or the origin of the sample (Steinke et al., 2020), there is currently no simple procedure to identify and quantify subpopulations or classes from cumulative freezing spectra $N_m(T)$. The second aim of our study is to map the cumulative freezing spectrum $N_m(T)$ into the differential spectrum $n_m(T)$, in terms of subpopulations that may correspond to different physical nucleation sites in the sample.

To reach the aims above, we develop a method we name HUB (for Heterogeneous Underlying-Based) to model and interpret the results of drop-freezing experiments and provide its associated Python code and user manual (https://github.com/Molinero-Group/underlying-distribution). Our method relies on the singular interpretation of freezing experiments: we assume that each individual IN has a characteristic nucleation temperature independent of its cooling history, and that the freezing of a droplet containing multiple INs is promoted by the IN with the highest nucleation temperature. This second assumption allows the use of extreme value statistics (Castillo, 2005; David and Nagaraja, 2004; Gumbel, 2012; De Haan and Ferreira, 2006) to model and interpret the data.

We present two implementations of the HUB analysis code. The HUB-forward code allows the user to postulate an underlying distribution of heterogeneous nucleation temperatures $P_u(T)$ in the system of interest. The HUB-forward code uses the singular approximation and extreme-value statistics to generate an artificial IN dilution series similar to those obtained in experiments, from which it computes the fraction of frozen droplets $f_{ice}(T)$ and from these derive $N_m(T)$ using Vali's equation (**Fig. 1**). The HUB-backward code works in reverse, extracting the differential spectrum $n_m(T)$ from a given cumulative $N_m(T)$ using a stochastic optimization procedure (**Fig. 1**). HUB-backward allows the decomposition the total population from $n_m(T)$ into subpopulations. The combination of HUB-forward and HUB-backward allows for an analysis of the sensitivity of $N_m(T)$ to the number of droplets and dilutions, and the impact of the sampling on the closeness of the differential spectrum $n_m(T)$ to the underlying distribution $P_u(T)$. The determination of distributions obtained from the HUB-backward code could further enable the interpretation of the experimental ice nucleation spectra with size and structure of INs using nucleation theory, kinetic models, and molecular simulations. For example, (Schwidetzky et al., 2023) illustrates the use of the distribution of freezing temperatures obtained with HUB-backward together with classical nucleation theory for finite surfaces to interpret the size of the IN of *Fusarium acuminatum*.

This paper is organized as follows: **Section 2** presents the methodology: **Section 2.1** discusses the details on the implementation of HUB-forward, while **Section 2.2** describes the HUB-backward procedure to find the differential spectrum $n_m(T)$ and discusses how to determine whether or not $n_m(T)$ has converged to the underlying distribution $P_u(T)$. **Section 3** presents examples of applications of both HUB-forward and HUB-backward codes and their capabilities. **Section 3.1** analyses the effect of the number of droplets sampled on the cumulative freezing spectrum $N_m(T)$. **Section 3.2** uses HUB-backward to compute the differential spectra $n_m(T)$ of various biological INs with increasing grade of complexity in their

cumulative freezing spectra. **Section 3.3** demonstrates how to extract $n_m(T)$ from the experimental fraction of ice $f_{ice}(T)$ of insoluble INs and the impact of the cooling rate on $n_m(T)$. We end in **Section 4** with a discussion of the main conclusions and outlook.

## 2. Numerical modeling of drop-freezing experiments

### 2.1 HUB-forward method to compute the fraction of frozen droplets $f_{ice}(T)$ and cumulative freezing spectrum $N_m(T)$ from a known underlying distribution $P_u(T)$.

In the HUB-forward analysis we know or assume an underlying distribution $P_u(T)$ of ice nucleation temperatures for the IN in the sample, and generate from it an artificial IN dilution series similar to those obtained in experiments, from which we compute the cumulative freezing spectrum $N_m(T)$ using Vali's equation (**Eq. (1a)**). Using this approach, we investigate the relationship between $N_m(T)$ and $P_u(T)$ (**Fig. 1**) and the sensitivity of $N_m(T)$ with respect to the number of droplets and dilutions. For generality, we represent $P_u(T)$ as a linear combination of normalized continuous distributions $P_i(T)$ that represent subpopulations of freezing temperatures:

$$P_u(T) = c_1 P_1(T) + c_2 P_2(T) + \ldots + c_p P_p(T), \tag{2}$$

where $p$ is the total number of subpopulations, $P_1(T), P_2(T), \ldots, P_p(T)$ are normalized distribution functions, and $c_1, c_2, \ldots, c_p$ are their weights such that $\sum_{i=1}^{p} c_i = 1$. These subpopulations could correspond to different chemical, topographical or structural motifs in the IN samples, although chemically distinct species could also produce overlapped freezing signatures, and a single species could display a broad freezing range. Our formalism does not require a mapping of subpopulations of freezing temperatures to physical IN sites. The units of $P_u(T)$ are, same as for $n_m(T)$, i.e. those of the cumulative spectrum divided by a unit of temperature, but are generally omitted in what follows. Throughout this work we assume that $P_i(T)$ can be represented by Gaussian (i.e. normal) distributions:

$$P_i(T) = \left(\frac{1}{s_i\sqrt{2\pi}}\right) e^{-\frac{1}{2}\left[\frac{T - T_{mode,i}}{s_i}\right]^2}, \tag{3}$$

where each subpopulation $P_i(T)$ is further characterized by its most likely temperature of freezing $T_{mode,i}$ and spread of distribution of freezing temperatures $s_i$. We also provide in the HUB code the option for the user to use the log-normal distribution, which has a tail towards higher temperatures, or the left-tailed Gumbel distribution, which has a tail towards lower temperatures. In our model, we assume that the underlying distribution of ice nucleating temperatures $P_u(T)$ does not change with the concentration of INs. This last condition is violated when IN are involved in chemical, aggregation, or solubility equilibria that alter the proportionality between their concentration and the dilution factor of the sample, resulting in a lack of overlap of the pieces of the cumulative spectra $N_m(T)$ obtained from different dilutions (Bogler and Borduas-Dedekind, 2020)

The number of INs in each droplet is then given by the Poisson distribution:

$$p(n, \lambda) = \frac{\lambda^n}{n!} e^{-\lambda}, \tag{4}$$

where $n$ is the actual number of INs in each droplet and $\lambda$ represents the average number of INs among all droplets of the corresponding dilution. **Fig. 2A** shows the probability mass function (PMF) for $\lambda = 1, 5$, and 10, computed according to **Eq. (4)** and sampling over $N_0 = 10^4$ droplets using the "SciPy Stats" Python framework (Virtanen et al., 2020). As $\lambda$ increases, the probability that any droplet nucleates homogeneously rapidly approaches zero (inset of **Fig. 2A**). When there is one IN on average per droplet ($\lambda = 1$) ~37 % of the droplets do not have any IN, i.e., they are "empty" droplets that would nucleate at the homogeneous nucleation temperature. We note that by performing dilutions until a sizeable fraction of droplets nucleate homogeneously, it is possible to calibrate the absolute concentration of ice nuclei in the original, undiluted, sample.

To illustrate how the heterogeneous ice nucleation temperatures recorded in drop-freezing experiments depend on the number of INs in the droplets, we start from two examples with $P_u(T)$ represented by one or two Gaussian subpopulations, shown with black dashed lines in **Fig. 2B** and **C**, respectively. We assign a temperature to each IN contained in droplets from a 10-fold dilution series of 5 solutions with $\lambda = 1, 10, 10^2, 10^3$, and $10^4$ average number of IN per droplet. If the droplet volume is constant, $\lambda$ is proportional to the concentration of INs in the droplets. We sample $N_0 = 10^4$ droplets for each concentration. This $N_0$ is much higher than the ~100 droplets usually sampled in laboratory experiments; we address the effect of sampling in **Section 3.1** below.

To sample independent random values for each IN, the number of random variates, which are drawn from $P_u(T)$, is the total number of INs among $N_0$ droplets. Thereby, each droplet has a set of temperatures $T_j^\lambda = (T_1^\lambda, T_2^\lambda, \ldots, T_k^\lambda)$ where $j$ is the droplet index and $k$ is the IN index. Since we assume that freezing occurs at the characteristic temperature of the IN with the highest freezing temperature, the nucleation temperature for each droplet is defined as the maximum i.e., the extreme upper value, of several independent freezing temperatures $T_{het,j}^\lambda = max(T_1^\lambda, T_2^\lambda, \ldots, T_k^\lambda)$. **Fig. 2B-C** shows the normalized distribution of $T_{het,j}^\lambda$ for different values of $\lambda$, namely $P_{max}^\lambda(T)$. Therefore, $P_u(T)$ represents the underlying probability of heterogeneous ice nucleation temperatures independent of the concentration of INs, while $P_{max}^\lambda(T)$ represents the concentration-dependent distribution, and has the same units as $P_u(T)$ and the differential spectrum. According to the Fisher-Tippett-Gnedenko theorem, the distribution of extreme upper values of the Gaussian distribution is the right-skewed Gumbel distribution (Castillo, 2005; David and Nagaraja, 2004; Gumbel, 2012; De Haan and Ferreira, 2006), which has a fatter tail on the high-temperature side of its maximum. The shift of $P_{max}^\lambda(T)$ curves in **Fig. 1B-C** evinces that as the number of INs in the droplet increases, the probability of sampling the higher temperature tail of $P_u(T)$ increases significantly. This skew is the reason why several dilutions are needed to sample the full population of ice nucleants.

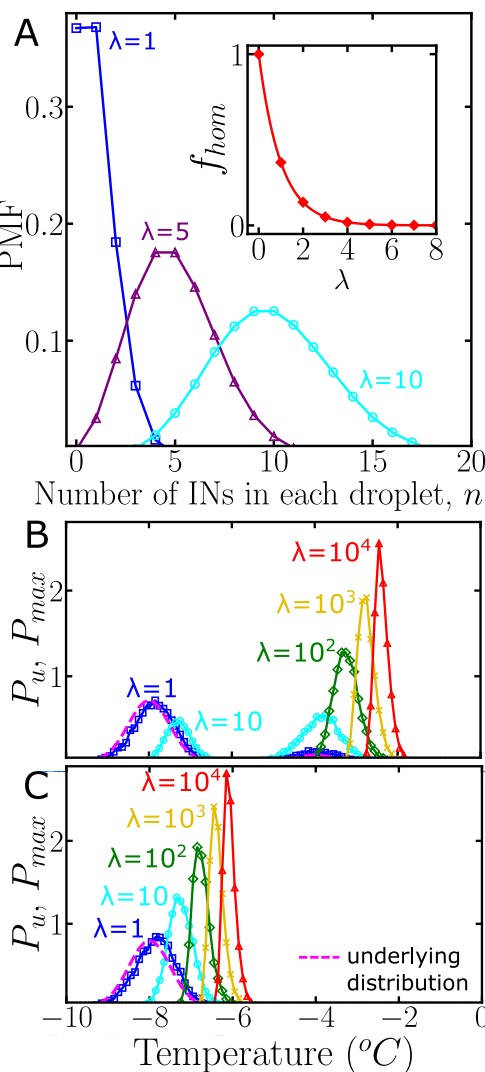

**Figure 2: A) The probability mass function (PMF) of the Poisson distribution representing the number of INs per droplet using a bin width of 1. Colors represent different average numbers of INs per droplet: $\lambda = 1$ (blue squares), $\lambda = 5$ (purple triangles) and $\lambda = 10$ (cyan circles). The inset shows the fraction of empty droplets as a function of $\lambda$. The connecting lines are solely guides to the eye. B) and C) show the normalized underlying distributions $P_u(T)$ of heterogeneous ice nucleation temperatures (magenta dashed line), composed of two subpopulation and one subpopulations, respectively. Colors represent the concentration-dependent normalized distribution $P_{max}^\lambda(T)$ of heterogeneous ice nucleation temperatures: $\lambda = 1$ (blue squares), $\lambda = 10$ (cyan circles), $\lambda = 10^2$ (green diamonds), $\lambda = 10^3$ (yellow x) and $\lambda = 10^4$ (red triangles) INs per droplet. A bin width of 0.1 was used for $P_u(T)$ and $P_{max}^\lambda(T)$. All distributions were obtained using $10^4$ droplets. While the HUB-Forward code explicitly accounts for $N_F^\lambda$ and $N_0$, we note that their ratio can be approximated by $N_F^\lambda/N_0 \approx (1 - e^{-\lambda})$ based on properties of the Poisson distribution.**

HUB-forward computes the fraction of frozen droplets and cumulative spectra from a proposed underlying distribution of freezing temperatures, using extreme-value statistics. The fraction of frozen droplets $f_{ice}^\lambda(T)$ can be calculated as a function of the concentration-dependent distribution,

$$f_{ice}^\lambda(T) = \int_{T_m}^{T} P_{max}^\lambda(T')dT' \times \frac{N_F^\lambda}{N_0}, \tag{5}$$

where the integration is from the ice melting temperature $T_m$ to the temperature $T$. $N_F^\lambda$ is the total number of droplets that freeze heterogeneously and $N_0$ is the total number of droplets. We note that the approach taken in this work differs from that of previous studies that either start from a microscopic model for the nucleation sites and nucleation theory to predict the fraction of frozen droplets using Monte Carlo simulations, and also from previous modelling using the singular
approximation which do not account for the statistics of extreme sampling.

To use the HUB-Forward code, the user must define the total number of droplets "ndroplets" that serves as the total number of each concentration and the number of subpopulations "nsubpop". If "nsubpop"=1, the user must provide the temperature of maximum likelihood $T_{mode,1}$ and the spread $s_1$. If "nsubpop"=2, the user must provide $T_{mode,1}, s_1, T_{mode,2}$, $s_2$ and $c_2$. If "nsubpop"=3, the user has to provide $T_{mode,1}, s_1, T_{mode,2}, s_2, c_2, T_{mode,3}, s_3, c_3$. To generate the cumulative
freezing spectrum $N_m(T)$, the user needs to define the total number of concentrations "nconc", the concentration of the parent suspension is defined in "density", and the droplet volume in "volumedrop". The output is composed of different data plots and files: the normalized $P_u(T)$ and $P_{max}^\lambda(T)$, the artificially generated $f_{ice}^\lambda(T)$, and $N_m(T)$ built from the 10-fold dilution series.

**Fig. 3A-B** shows the fraction of ice computed using $P_{max}^\lambda(T)$ of **Fig. 2B-C**, which correspond to $P_u(T)$ with two
and one subpopulations, respectively. The intermediate plateau in **Fig. 3B** indicates that no droplets freeze at those temperatures. As discussed above, only 63% of the droplets freeze heterogeneously for $\lambda = 1$. We assume droplets of uniform volume $V_{drop} = 0.1$ µl obtained through 10-fold dilution of a parent suspension with $\lambda = 10^4$ INs per droplet corresponding to a mass $m = 1$ mg of IN in a volume $V_{wash} = 1$ ml. We use **Eq. (5)** and the $P_u(T)$ of Fig. 2B-C to generate $f_{ice}^\lambda(T)$ (**Fig. 3A-B**) sampling of either 100 or $10^4$ droplets per dilution. We combine the $f_{ice}^\lambda(T)$ using **Eq. (1a)** to build the
cumulative freezing spectra $N_m(T)$ shown in **Fig. 3C-D** (sampling $10^4$ droplets per dilution) and **Fig. 3E-F** (sampling $10^2$ droplets per dilution).

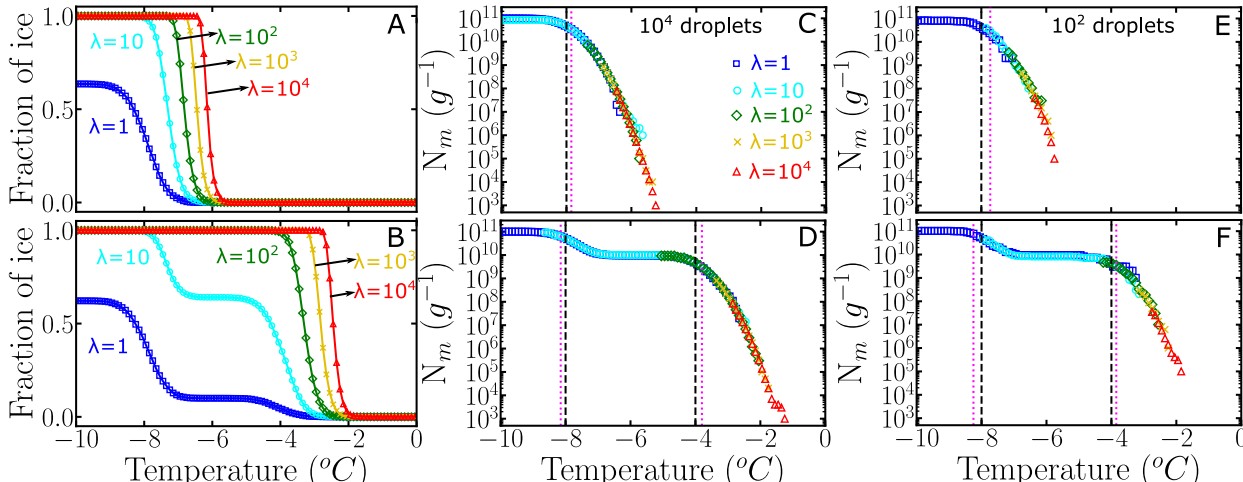

**Figure 3: A) and B) represent the fraction of ice $f_{ice}^{\lambda}(T)$ computed using Eq. (5) and artificially generated data using $10^4$ droplets. C) and D) are the corresponding cumulative freezing spectra $N_m(T)$ computed using Vali's equation. Colors represent different number of INs per droplet: $\lambda = 1$ (blue squares), $\lambda = 10$ (cyan circles), $\lambda = 10^2$ (green diamonds), $\lambda = 10^3$ (yellow x) and $\lambda = 10^4$ (red triangles). E) and F) represent $N_m(T)$ obtained using 100 droplets. The dashed black lines in C and D indicate the temperatures corresponding to the location of the mode(s) in the underlying distribution. The dotted magenta lines are the knee points computed with the Python function "kneed".**

The ability of HUB-forward to generate the cumulative freezing spectrum $N_m(T)$ from the underlying distribution $P_u(T)$ allows for an analysis of the sensitivity of $N_m(T)$ and $P_u(T)$ to the number of droplets and dilutions, as seen in the comparison of $N_m(T)$ generated from the same underlying distributions using 100 and $10^4$ droplets in **Fig. 3**. In **Section 3.1** we show that the sampling with 100 droplets for only four dilutions of a system with two subpopulations of INs results in distortions of the freezing temperatures and the proportions of these populations in the differential spectrum.

The knee point in $N_m(T)$ corresponds to the point of maximum curvature (Satopaa et al., 2011) and has been used to characterize the nucleation temperature of a particular subpopulation (Hartmann et al., 2022). Similar to Hartmann *et al.*, we have identified in **Fig. 3C-D** the knee points (magenta dotted line) of the artificially generated $N_m(T)$ by using a Python function named "kneed". The Python function "kneed" using S=1, curve="concave" and direction="decreasing". The knee points $T_{knee}$ are very close to the temperatures of maximum likelihood $T_{mode}$ (dashed black lines) of the corresponding underlying distribution $P_u(T)$, because under these conditions the differential freezing spectrum $n_m(T)$ is a very good approximant for $P_u(T)$. However, we find that removal of the more dilute solutions (that eliminate the plateau in $N_m(T)$) results in poor estimation of the modes of $P_u(T)$ from the knee of $N_m(T)$.

**2.2. HUB-Backward method to recover the differential freezing spectrum $n_m(T)$ from the cumulative freezing spectrum $N_m(T)$ by a stochastic optimization procedure.**

The HUB-backward code implements a stochastic optimization procedure to extract the differential spectrum $n_m(T)$ from a given cumulative spectrum $N_m(T)$ or from an experimental $f_{ice}(T)$ curve. The later is useful when data is available for a single concentration. One possibility to obtain $n_m(T)$ from $N_m(T)$ would be to follow the following steps: i) propose a trial

function $n_m^{trial}(T)$, ii) use HUB-forward to predict the concentration-dependent distributions $P_{max}^{\lambda,trial}(T)$ for various IN concentrations, iii) use these in **Eq. (5)** to predict the freezing fractions $f_{ice}^{\lambda,trial}(T)$, iii) compute $N_m^{trial}(T)$ from the freezing fractions using **Eq. (1a)**, iv) evaluate the difference between that trial and the target (experimental) value,

$$\delta(T) = \left| log_{10}[N_m^{trial}(T)] - log_{10}[N_m^{target}(T)] \right|, \tag{6}$$

and then v) evolve the parameters that determine $n_m^{trial}(T)$ until the difference $\delta(T)$ is minimized. However, the use of HUB-forward in steps ii) and iii) to generate and evaluate hundreds of droplets containing up to tens of millions of IN would require significant computations that render this optimization process inefficient.

The HUB-backward optimization procedure, sketched in **Fig. 4**, uses a shortcut for steps ii) and iii) above to directly predict $N_m^{trial}(T)$ from $n_m^{trial}(T)$ with fast convergence. The shortcut is based on the understanding that in the asymptotic limit in which the sample is extremely dilute (i.e., $\lambda \rightarrow 0$) each droplet that nucleates heterogeneously contains a single IN. In such case, sampling an infinitely large number of droplets with $P_{max}^{\lambda\rightarrow0}(T)$ is equivalent to sampling each and every IN, *i.e.*, $P_{max}^{\lambda\rightarrow0}(T) = P_u(T)$. In agreement with this ansatz, **Fig. 2B-C** shows that the underlying distribution $P_u(T)$ (black dashed line) and the concentration-dependent $P_{max}^{\lambda=1}(T)$ (blue squares) sampled with $10^4$ droplets per dilution are already very close, i.e. $P_u(T) \approx P_{max}^{\lambda=1}(T)$.

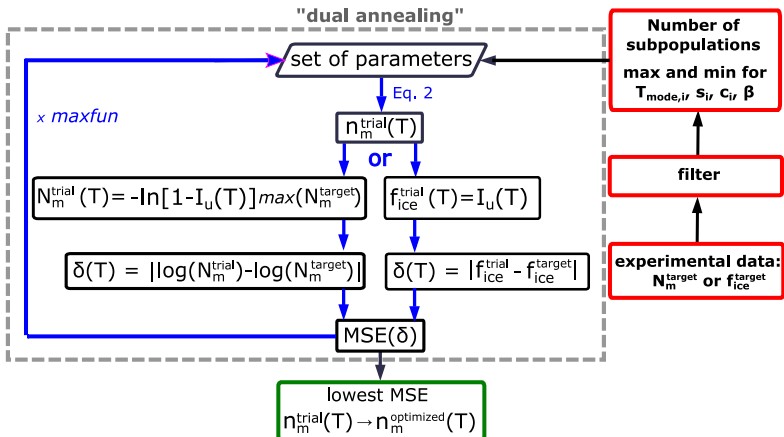

**Figure 4: Flowchart of the minimization procedure to obtain the differential freezing spectrum $n_m(T)$ from the full cumulative freezing spectrum $N_m(T)$ or fraction of frozen droplets $f_{ice}(T)$.**

With this insight and considering that the intrinsic cumulative spectrum, $I_u(T) = \int_{T_m}^{T} P_u(T')dT' \times (1 - e^{-1})$, we define the cumulative integral of the differential spectrum as

$$I_u^{trial}(T) = \int_{T_m}^{T} n_m^{trial}(T')dT' \times \beta, \tag{7}$$

where the integration is from the ice melting temperature $T_m$ to the temperature $T$, and $\beta$ is and adjustable scaling factor to be obtained from the optimization. Likewise, a similar estimate can be made for a single fraction of ice curve $f_{ice}^{trial}(T) = I_u^{trial}(T)$ using **Eq. (7)** and proceed directly to evaluate the mean squared error (**Fig. 4**). When the target is a cumulative freezing spectrum, HUB-forward uses $n_m^{trial}(T)$ to predict a trial cumulative freezing spectrum (**Fig. 4**),

$$N_m^{trial}(T) = -\ln{[1 - I_u(T)]} \times \frac{1}{X}, \tag{8}$$

where $1/X$ corresponds to the maximum of the cumulative in the target distribution, $1/X = \max\left[N_m^{target}(T)\right]$. With **Eq. (8)** we obtain an $N_m^{trial}(T)$ that we compare with the target using **Eq. (6)** (**Fig. 4**). To do the comparison, HUB-backward uses a spline fit to interpolate the experimental $N_m^{target}(T)$ in order to have equally spaced temperature points to then compare with the estimates in $N_m^{trial}(T)$. We use the "interp1d" algorithm, which is available in the Python SciPy library (Virtanen et al., 2020) with a linear interpolation to construct new equally spaced data points within the range of the lowest and highest temperature values in the freezing spectrum. The cost function for the optimization is the mean squared error (MSE), computed from the difference $\delta(T)$ in **Eq. (6)**,

$$MSE = \frac{1}{t}\sum{\delta^2}, \tag{9}$$

where $t$ represents the total number of equally spaced points in $\delta(T)$.

We use a stochastic global optimization technique based on a simulated annealing algorithm to find the set of parameters of $n_m^{trial}$ (**Eq. (2)** and **Eq. (3)**), and $\beta$ (**Eq. (7)**) that globally minimizes the MSE. We use the simulated annealing (SA) algorithm "dual annealing" that is part of the SciPy minimize library (Virtanen et al., 2020) with its default arguments predefined, except for the parameters "maxfun" that sets the maximum number of the evaluation of the objective function (we select "maxfun" = 1,000,000 in the examples below), and the seed for the generation of random numbers (a new random integer is automatically generated every time the HUB-backward code is run). We show below that the optimized differential spectra, $n_m^{optimized}(T)$, are quite insensitive to the value of the seed.

The output of HUB-backward is an optimized differential spectrum $n_m^{optimized}(T)$ or an optimized fraction of ice $f_{ice}^{optimized}(T)$. To quantify how much this optimized prediction deviates from the known underlying distribution in the examples of **Fig. 5** where $P_u(T)$ is known, we define the mean relative error (MRE) for the set of parameters

$$MRE = \frac{1}{3p}\sum_{i=1}^{p}\left[\left|\frac{T_{mode,i}^{optimized} - T_{mode,i}^{target}}{T_{mode,i}^{target}}\right| + \left|\frac{s_i^{optimized} - s_i^{target}}{s_i^{target}}\right| + \left|\frac{c_i^{optimized} - c_i^{target}}{c_i^{target}}\right|\right], \tag{10}$$

where $p$ is the number of subpopulations.

We now turn our focus to how to select the input parameters required by HUB-backward to start the search for the underlying distribution, using the experimental $N_m^{target}(T)$ or $f_{ice}^{target}(T)$ as a guide. The code requires the user to define the

number of distinct Gaussian subpopulations $P_i(T)$ that comprise the underlying distribution (**Eq. (2)**) and to provide upper and lower bounds for the weighs $c_i$, their modes $T_{mode,i}$, and spreads $s_i$ of each of these populations. In general, we find that defining the minimum and maximum values for the weighs $c_i^{max} = 1$ and $c_i^{min} = 0$ (see constrain in **Eq. (2)**), $T_{mode,i}^{max}$ and $T_{mode,i}^{min}$ to be between the homogeneous nucleation temperature (about -30 °C) and the melting temperature (0 °C), and the bounds for the spreads $s_i^{max} = 10$ °C and $s_i^{min} = 0.1$ °C work well. However, these bounds can be tuned in order to better fit the data (as we find to be the case to fit the results for pollen in **Section 3.2** below). If the existing experimental $N_m^{target}(T)$ data is very noisy, it can be interpolated in HUB-backward using the method "interp1d" with "npoints"=100, and then smoothed with a Savitzky-Golay filter by changing the parameters "window_length" that is the length of the filter window and "polyorder" that is the order of the polynomial used to fit the samples ("filter" in **Fig. 4**). The default values are 3 and 1, respectively. HUB-backward generates a plot that compares the original and the interpolated target data.

To identify the minimum number of subpopulations needed to represent a given freezing spectrum, we consider that every time a population is accumulated in $N_m(T)$ or $f_{ice}(T)$, these functions display a sharp increase. We note that assuming a large number of subpopulations may challenge the interpretability of the optimized differential spectrum $n_m^{optimized}(T)$.

We apply the HUB-backward procedure to the $N_m(T)$ obtained in **Fig. 3C-D** by sampling four 10-fold dilutions with 100 droplets, i.e., only a total of 500 droplets. **Fig. 5** shows the comparison between the predicted (solid magenta lines) and the target (black dashed lines) $N_m(T)$ (panels **A** and **B**) and $n_m(T)$ (panels **C** and **D**). **Table 1** shows the predicted parameters and the precision of the optimization procedure to recover the known underlying distribution $P_u(T)$. The MRE between the underlying distribution $P_u(T)$ and the optimized differential spectrum $n_m^{optimized}(T)$ is 2% for the system with one subpopulation and 13% for the one with two, despite the low number of droplets used to sample the cumulative freezing spectra in the computer-generated freezing experiments.

**Table 1: Mean relative error (MRE), mean squared error (MSE) and parameters of the optimized differential freezing spectra $n_m^{optimized}(T)$ obtained using the HUB-backward code. The values shown here were calculated based on the average of $n = 3$ independent runs. The error bars, shown in parentheses, were calculated by dividing the standard deviation of the values in these runs by $3^{1/2}$.**

| | MRE | MSE | $T_{mode,1}$ (·C) | $s_1$(·C) | $T_{mode,2}$ (·C) | $s_2$(·C) | $c_2$ | $\beta$ |
|---|---|---|---|---|---|---|---|---|
| **One subpopulation** | 2% | $1.0(2) \times 10^{-3}$ | -7.80(2) | 0.49(2) | | | | 0.63(1) |
| **Two subpopulations** | 13% | $3.0(2) \times 10^{-3}$ | -7.90(2) | 0.54(2) | -3.90(2) | 0.49(2) | 0.16(2) | 0.63(1) |

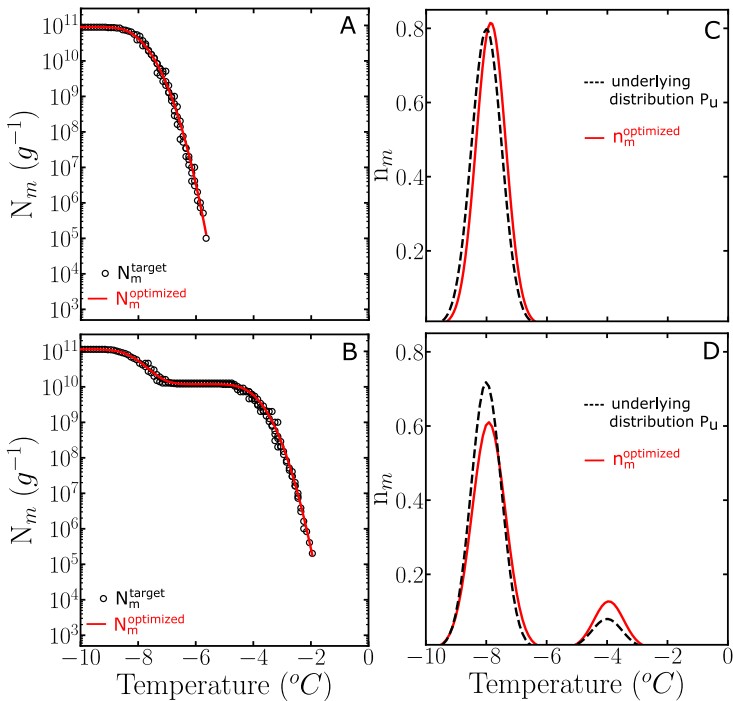

 **Figure 5: A) and B) show the comparison between $N_m^{target}(T)$ (black circles) and the $N_m^{optimized}(T)$ computed with the optimized solution $n_m^{optimized}(T)$ (solid red line). C) and D) show the known underlying distributions $P_u(T)$ (black dashed line) and the optimized underlying distributions $n_m^{optimized}(T)$ (solid red line) based on three independent runs. The parameters of the predicted underlying distribution $n_m^{optimized}(T)$ are summarized in Table 1.**

We conclude that the HUB-backward code gives a good estimate of the mode, spread and weights of the populations of INs in a sample, and it can be applied in a situation where $P_u(T)$ is unknown. In **Section 3.1** we discuss how is the accuracy of the underlying distribution recovered with HUB-backward impacted by various schemes of sampling of number of droplets and dilutions to construct $N_m(T)$. In **Section 3.2**, we apply the HUB-backward procedure to obtain $n_m^{optimized}(T)$ from actual $N_m(T)$ of experiments with various soluble biological IN. In **Section 3.3**, we apply the HUB-backward procedure to obtain $n_m^{optimized}(T)$ from $f_{ice}(T)$ of experiments of insoluble crystal IN.

**3. Using the HUB code to optimize and analyse drop-freezing experiments**

**3.1 Effect of the number of droplets and dilutions on the temperature range of the cumulative freezing spectrum $N_m(T)$**

**Fig. 3D-F** shows $N_m(T)$ generated with HUB-forward using five dilutions from $\lambda = 10^4$ to 1 of a solution with $P_u(T)$ containing two populations in a ratio of 9 to 1. The $N_m(T)$ are different when the number of droplets per dilution is 100 (**Fig. 3F**) or $10^4$ (**Fig. 3D**). As shown in the previous section, the freezing spectrum obtained with 100 droplets and five dilutions

has enough sampling to recover this $P_u(T)$ with good accuracy (**Fig. 5C-D**). We test different number of droplets and concentrations, defined by the average number of INs per droplet $\lambda$, to test the sensitivity of $n_m(T)$ to the number of droplets and dilutions when the underlying distribution is known $P_u(T)$. We use HUB-forward to build $N_m(T)$ based on a combination of different number of droplets and concentrations, similar to the case shown in **Fig. 3F**. Then, we use the HUB-backward to obtain $n_m^{optimized}(T)$, compare it to $P_u(T)$ and test the accuracy of each prediction through its mean relative error (MRE) as defined in **Eq. (10)**.

      The left panels of **Fig. 6** show $N_m(T)$ generated with HUB-forward based on a combination of different concentrations using 100 droplets each. The magenta lines are based on the data fitting provided by the HUB-backward code. The right panels of **Fig. 6** compare $n_m^{optimized}(T)$ in magenta and the known underlying distribution $P_u(T)$ in black. In this example, $n_m(T)$ is very close to $P_u(T)$ if both subpopulations are sampled enough. However, if the most dilute solution with $\lambda = 1$ is not included in $N_m(T)$ (second panel), the estimate of the underlying distribution is very poor. Thus, to improve the sampling of the lower tail of $P_u(T)$, we recommend ending the dilution series always in the immediacy of $\lambda = 1$, which can be gleaned from the temperature range for which $N_m(T)$ becomes flat and a sizeable fraction of droplets of the more diluted sample nucleates homogeneously (inset of **Fig. 2A**). We emphasize that reaching this limit allows for an *absolute* calibration of the number of INs in the initial sample. Moreover, sampling to concentrations down to about one nucleant per droplet is essential to recover a proper weight of the poorly nucleating IN populations.

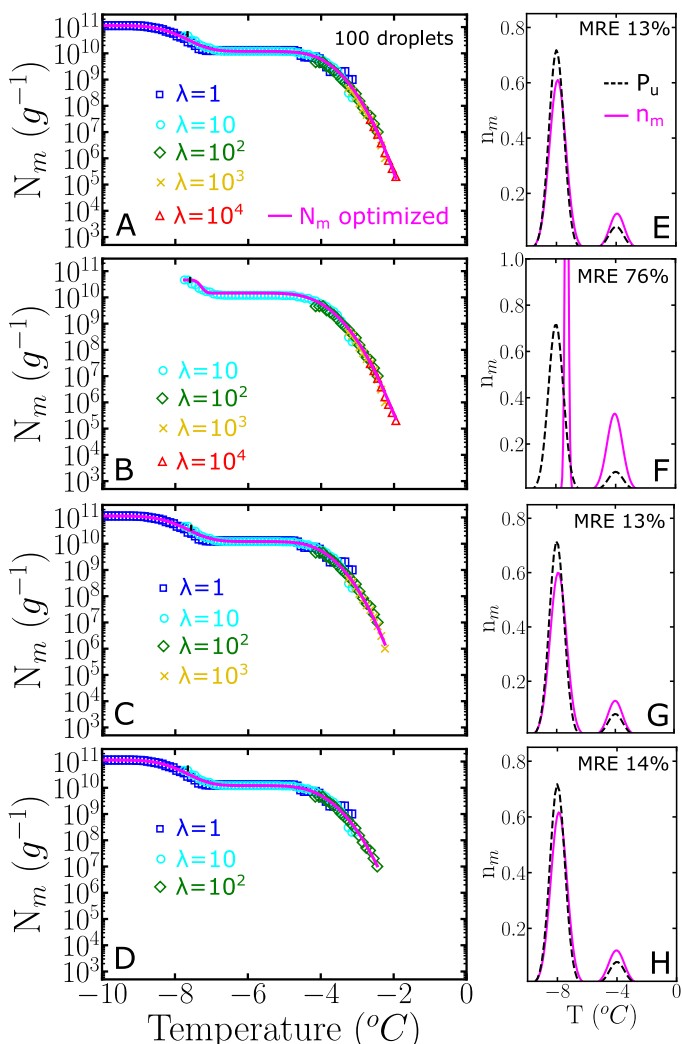

**Figure 6: Left panels represent the cumulative freezing spectra $N_m(T)$ sampled from the same underlying distribution $P_u(T)$.** Colors represent different number of INs per droplet: $\lambda = 1$ (blue squares), $\lambda = 10$ (cyan circles), $\lambda = 10^2$ (green diamonds), $\lambda = 10^3$ (yellow x) and $\lambda = 10^4$ (red triangles). The sampling was done using 100 droplets for each concentration. Right panels represent the differential freezing spectra $n_m(T)$ compared to the known underlying distribution $P_u(T)$, shown by the magenta and dashed black lines, respectively. Panels A, B, C and D were computed with a different number of dilutions. The mean relative error (MRE) was computed using Eq. (10). The parameters of $n_m(T)$ and $P_u(T)$ are shown in Table S1.

The relative weights of class A and C populations in *Pseudomonas syringae* is approximately 1 to 1000 (**Section 3.2),** while the ratio is 9 to 1 in the two-population system example of **Fig. 6**. To understand the impact of highly imbalanced populations on the sampling of the cumulative spectrum and recovery of the underlying distribution, we show in **Fig. 7** the analysis of an example where the subpopulation of highly efficient INs is three orders of magnitude less likely to occur than the subpopulation at lower temperatures, mimicking the one of *P. syringae*. Our analysis confirms that it is important to end the dilution series in the immediacy of $\lambda = 1$ to fully represent the contribution of the poorer INs (**Fig. 7B-F**). Furthermore,

we find it is important to sample a high enough concentration to account for the rare INs that nucleate at the highest temperatures (**Fig. 7D-H**).

If only 25 droplets per dilution, instead of 100, are used to construct the cumulative spectrum, the impact of insufficient sampling at the higher concentrations is more pronounced: compare **Fig. 8C** and **Fig. 7D** obtained with the same underlying distribution $P_u(T)$ with 1000 to 1 subpopulation ratios and number of dilutions.

We conclude that increase in the accuracy in the account of the subpopulations requires a higher number of dilutions and the checking of the predictions with the addition of each successive concentration, to ensure convergence of $n_m^{optimized}(T)$. Measuring fewer droplets or fewer dilutions lead to poor statistics and results in incompleteness or the misrepresentation of the underlying distribution in samples with multiple subpopulations. In principle, increasing the number of droplets of the most concentrated solutions, or adding more ten-fold concentrated ones until there are no changes in the cumulative spectrum is recommended to ensure complete sampling. When that limiting scenario is not attainable, the use of HUB-forward to produce synthetic data from a proposed underlying distribution, followed by the recovery of the differential spectrum from these data sets, allows for an estimation of the errors that may be incurred for putative, proposed underlying distributions with the sampling scheme available in the laboratory.

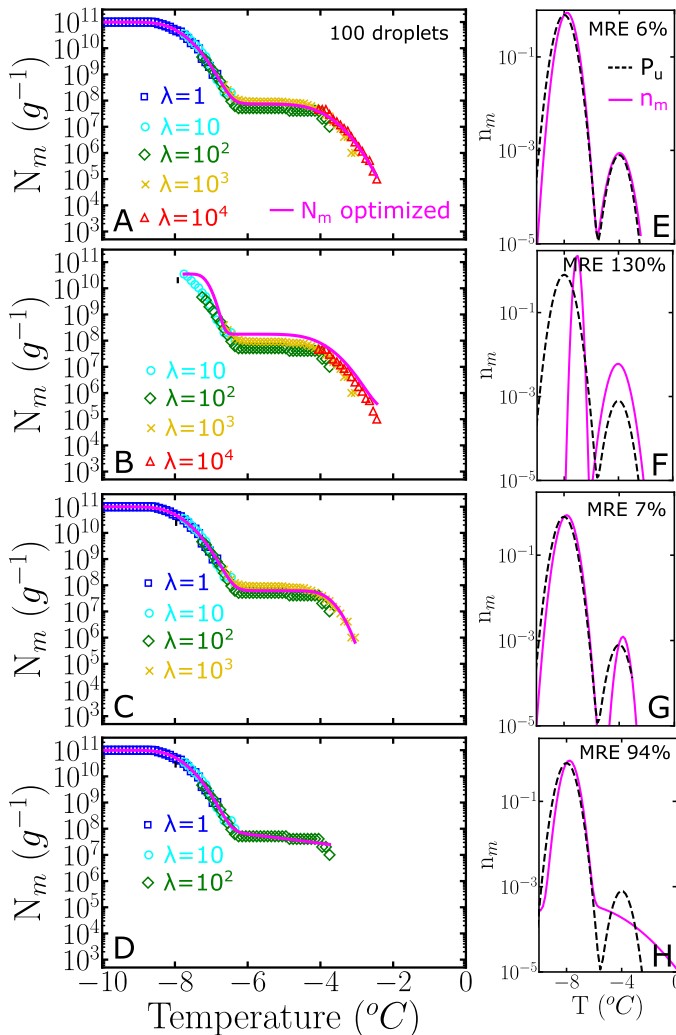

**Figure 7: Left panels (A-D) represent the cumulative freezing spectra $N_m(T)$ sampled from the same underlying distribution $P_u(T)$. Colors represent different number of INs per droplet: $\lambda = 1$ (blue squares), $\lambda = 10$ (cyan circles), $\lambda = 10^2$ (green diamonds), $\lambda = 10^3$ (yellow x) and $\lambda = 10^4$ (red triangles). The sampling was done using 100 droplets for each concentration. Right panels (E-H) represent the differential freezing spectra $n_m(T)$ compared to the known underlying distribution $P_u(T)$, shown by the magenta and dashed black lines, respectively. The mean relative error (MRE) was computed using Eq. (10) and the parameters of $n_m(T)$ and $P_u(T)$ are shown in Table S2.**

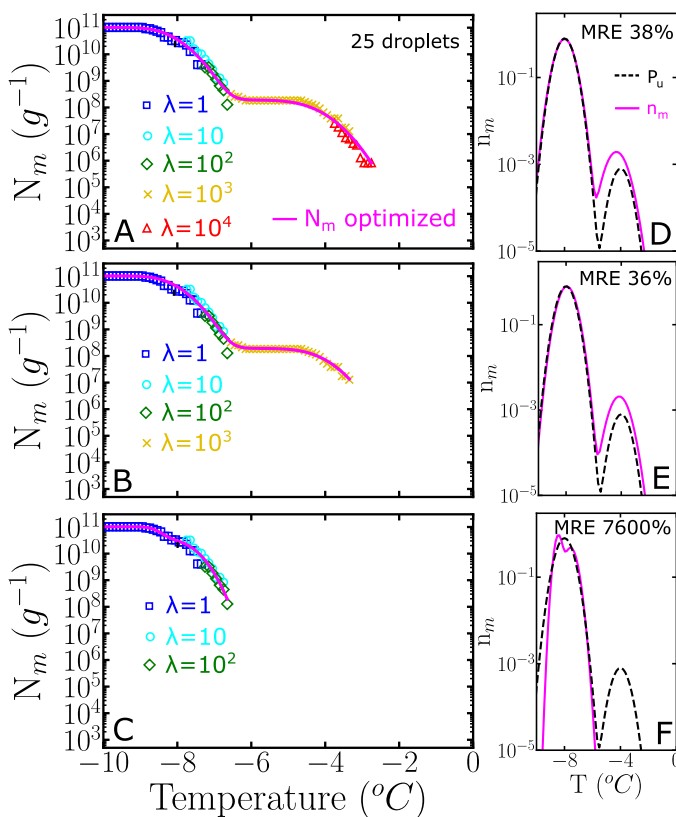

**Figure 8: Left panels (A-C) represent the cumulative freezing spectra $N_m(T)$ sampled from the same underlying distribution $P_u(T)$. Colors represent different number of INs per droplet: $\lambda = 1$ (blue squares), $\lambda = 10$ (cyan circles), $\lambda = 10^2$ (green diamonds), $\lambda = 10^3$ (yellow x) and $\lambda = 10^4$ (red triangles). The sampling was done using 25 droplets each concentration. Right panels (D-F) represent the differential freezing spectra $n_m(T)$ compared to the known underlying distribution $P_u(T)$, shown by the magenta full and black dashed lines, respectively. The mean relative error (MRE) was computed using Eq. (10). The parameters of $n_m(T)$ and $P_u(T)$ are shown in Table S3.**

### 3.2 Obtaining the differential freezing spectrum $n_m(T)$ from the experimental cumulative freezing spectrum $N_m(T)$ of biological INs using the HUB-backward code

In this section we use the HUB-backward code to obtain the differential freezing spectrum $n_m(T)$ from the cumulative freezing spectra $N_m(T)$ of the fungi *Fusarium acuminatum* strain 3-68 (Kunert et al., 2019), the bacterium *P. syringae* (Schwidetzky et al., 2021), and birch pollen (Dreischmeier, 2019). We select these systems because they are important biological INs and show increasing complexity in terms of the apparent number of underlying distributions that define their freezing spectra.

The experimental $N_m(T)$ obtained for *F. acuminatum* (black squares in **Fig. 9A**) was obtained by sampling six 10-fold dilutions, each with 96 droplets (Kunert et al., 2019). **Fig. 9A** shows the cumulative spectra optimized assuming one (green curve) and two (cyan curve) subpopulations; **Fig. 9B** shows the corresponding optimized differential freezing spectra. The $n_m^{optimized}(T)$ with a single subpopulation that peaks at -5.9°C is unable to represent the cumulative density of the most

potent nuclei and misses the inflection around -5.9°C in the experimental data, resulting in a mean squared error MSE = 0.05. The $n_m^{optimized}(T)$ with two subpopulations has a lower MSE = 0.003 and a better fit that suggests a population that peaks at

435    -7.3°C and another at -5.5°C, in comparable amounts (**Table 2**). Most notably, the two subpopulations do not overlap in the differential freezing spectrum, supporting that they may indeed correspond to different physical entities. The improvement in the fit becomes apparent in the inset of **Fig. 9A**, which shows $N_m(T)$ on a linear scale. The significant slope of $N_m(T)$ even at the lowest temperatures indicates that sampling of more diluted solutions is needed to capture the contribution of the less active INs. An attempt to represent *F. acuminatum* nucleation data with three different subpopulations resulted in two of

them being almost identical. We conclude that adding a third subpopulation is unnecessary to reproduce the experimental cumulative freezing spectrum of *F. acuminatum*. We refer the reader to (Schwidetzky et al., 2023) for an interpretation of the size of the ice nucleating surface of *F. acuminatum* based on its differential spectrum and nucleation theory.

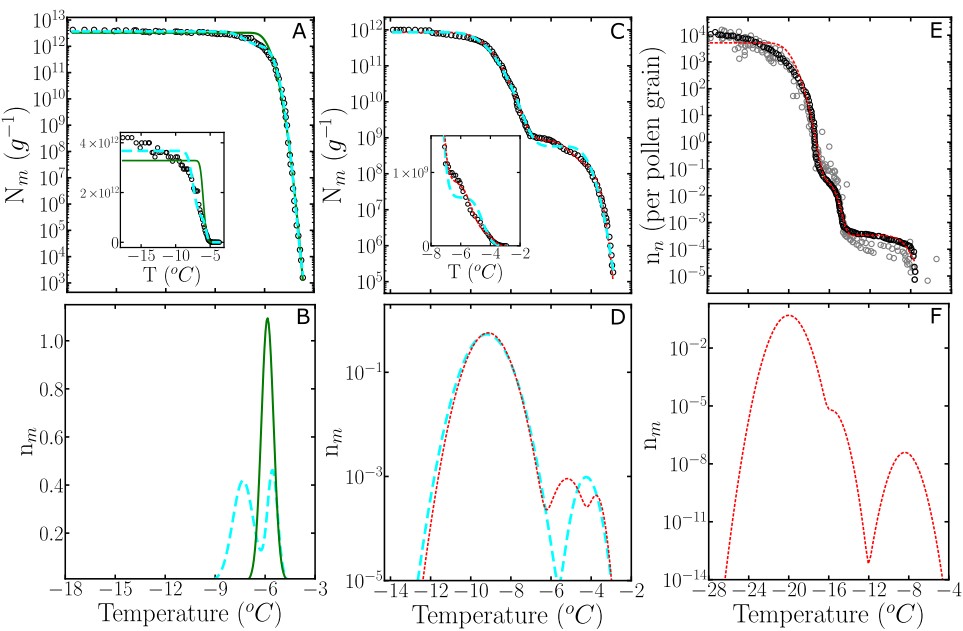

**Figure 9: Cumulative freezing spectra $N_m(T)$ obtained from drop-freezing experiments for A)** *F. acuminatum* **strain 3-68 (Kunert et al., 2019), B)** *P. Syringae* **(Schwidetzky et al., 2021), and C) birch pollen (Dreischmeier, 2019) (black circles). The solid green, long dashed cyan and short dashed red lines represent $N_m^{optimized}(T)$ computed with the optimized differential freezing spectra $n_m^{optimized}(T)$ obtained with the HUB-backward code considering one, two and three subpopulations, respectively. B), D) and E) show $n_m^{optimized}(T)$. The gray circles are experimental data points in the measurement of the birch pollen ice nucleation spectrum**

**that were not considered in the optimization procedure. Inset in A) and C) show $N_m(T)$ in normal scale.**

**Table 2: Mean squared error (MSE) and parameters of the differential freezing spectra $n_m(T)$ obtained using the HUB-backward code and experimental data as input. The values shown here were calculated based on the average of $n = 3$ independent runs. The error bars, shown in parentheses, were calculated by dividing the standard deviation of the values in these runs by $3^{1/2}$.**

| | Number of populations | MSE | $T_{mode,1}$ (°C) | $s_1$ (°C) | $T_{mode,2}$ (°C) | $s_2$ (°C) | $c_2$ | $T_{mode,3}$ (°C) | $s_3$ (°C) | $c_3$ | $\beta$ |
|---|---|---|---|---|---|---|---|---|---|---|---|
| *F. acuminatum* | 1 | 2.0% | -5.90(1) | 0.36(1) | | | | | | | 0.54(1) |
| *F. acuminatum* | 2 | 0.5% | -7.30(2) | 0.62(3) | -5.50(1) | 0.31(1) | 0.35(1) | | | | 0.58(2) |
| *P. syringae* | 2 | 2.0% | -9.40(2) | 0.77(2) | -4.20(2) | 0.41(3) | $7.0(2) \times 10^{-4}$ | | | | 0.87(1) |
| *P. syringae* | 3 | 1.1% | -9.10(2) | 0.70(2) | -5.20(1) | 0.53(2) | $1.0(1) \times 10^{-3}$ | -3.70(1) | 0.27(2) | $3.0(1) \times 10^{-4}$ | 0.57(1) |
| birch pollen | 3 | 5.0% | -20.00(2) | 0.79(3) | -15.60(2) | 0.58(1) | $9.0(1) \times 10^{-6}$ | -8.40(1) | 0.69(2) | $6.0(2) \times 10^{-8}$ | 0.39(3) |

Next, we apply the HUB-backward code to analyse the experimental freezing spectrum of Snomax®, *i.e.,* inactivated *P. syringae*. The cumulative spectrum suggests the presence of two distinct subpopulations, usually called class A (at warmer temperatures) and class C (at colder ones). We first assume the differential freezing spectrum $n_m(T)$ of *Ps. syringae* is a combination of two Gaussian populations. The parameters of the optimized differential spectrum with two subpopulations are listed in **Table 2**, and the curve is shown with in **Fig. 9D** with a cyan line. Note that we use a logarithmic scale to represent this $n_m^{optimized}(T)$ because the population corresponding to class A accounts for less than 0.1% of the total (**Table 2**). While the fit with two subpopulations results in a good overall account of the target data, we note that there is some difference in the region between classes A and C (**Fig. 9C**). The fitting for *P. syringae* achieves an excellent agreement between optimized and target cumulative spectra (**Fig. 9C**), through the prediction of an additional peak located between classes A and C (the elusive class B), with a population comparable to class A (**Table 2** and red curve in **Fig. 9D**). However, more measurements and analyses are needed to establish whether this "class B" peak at -5.2 °C is reproducible and truly distinct from the one of class A at -3.7 °C to warrant a physical interpretation. Overall, both the analyses with two and three subpopulations agree with previous ones (Govindarajan and Lindow, 1988; Warren, 1987) that concluded that over 99% of the IN active *P. syringae* bacteria in Snomax® belongs to class C. The analysis presented here for fungal and bacterial INs illustrates how HUB-backward can be used to reveal and characterize the underlying number of IN subpopulations of complex biological samples.

To further test the methodology, we model the cumulative freezing spectrum of birch pollen. Given that the original $N_m(T)$ data for pollen in Fig. 3.1 of (Dreischmeier, 2019) consists of multiple independent curves, we took one of the many presented in this graph as target $N_m^{target}(T)$ (black curve in **Fig. 9E**) and present some of the additional data -not used in the optimization- with gray circles in **Fig. 9E**. Supp. **Section S4** shows that the differential spectrum optimized from the whole data set and its sparse sampling are almost identical, because HUB-forward interpolates and smooths the input data to produce an equispaced data set. the $N_m^{target}(T)$ seems to contain three quite separated subpopulations; which is confirmed by the accuracy of the optimized cumulative spectrum in **Fig. 9E**. The parameters of the optimized differential freezing spectrum $n_m^{optimized}(T)$ and the MSE are shown in **Table 2**. Our analysis indicates that the two subpopulations that nucleate ice above -16°C constitute less than 0.01% of the active nucleating sites in pollen (**Fig. 9E**), consistent with drop-freezing assays that only measured solutions with low concentrations of birch pollen and did not observe freezing at higher temperatures (Augustin et al., 2013; Pummer et al., 2012; Felgitsch et al., 2018), where the more extensive data of (Dreischmeier, 2019) reveals two more active subpopulations of IN.

To further illustrate the use of HUB-backward, **Fig. 10** shows the effect of pH on the subpopulations in the modes, spread and weighs that contribute to the nucleation spectrum of *P. syringae* (Snomax®), using data from (Lukas et al., 2020). Freezing in the temperature range of class A drops about 3 orders of magnitude when the pH is lowered from 6.2 to 4.4, (**Fig. 10B**). However, we note that the cumulative number of IN is preserved in the experimental cumulative freezing spectrum (Lukas et al., 2020), indicating that the change in pH did not impact the number of nucleants. **Fig. 10C-D** demonstrates that the distributions associated with both subpopulations shift to lower temperatures when the pH decreases, and the range of freezing temperatures in class A becomes broader. An attempt to fit the cumulative spectra of Snomax at different pH with the same subpopulations, allowing only for adjustment of their weights, resulted in a poor fit to the experimental $N_m(T)$, supporting the conclusions of (Lukas et al., 2020) of a central role of electrostatic interactions in the assembly of the bacterial ice nucleating proteins and their ability to bind to ice. This analysis exemplifies how HUB-backward can be applied to quantify the dependence of IN on environmental variables.

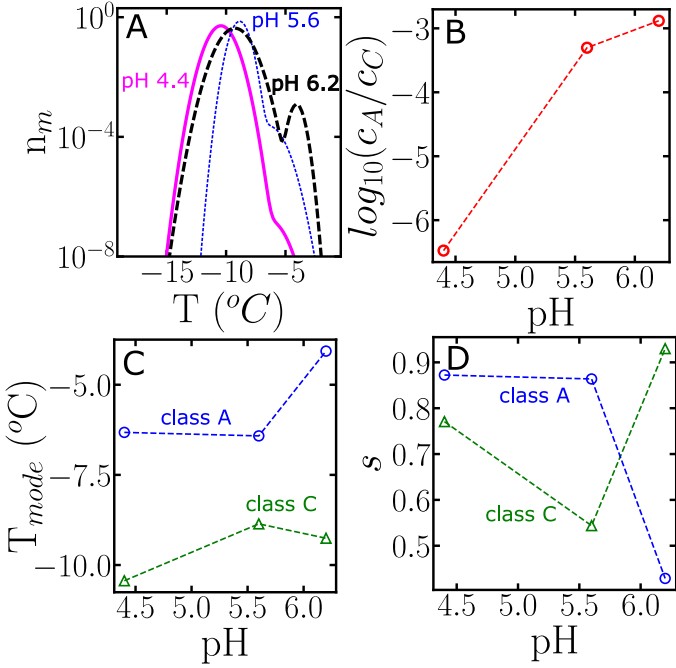

**Figure 10: Effect of changing the pH on the subpopulations of *P. Syringae* (Lukas et al., 2020). A) The differential freezing spectra $n_m(T)$ obtained using the HUB-backward code. Colors represent different pH: 6.5 (black long dashed line), 5.6 (blue short dotted line), and 4.4 (solid magenta line). B) The ratio between the weights, C) the modes, and D) the spreads of each subpopulation as a function of pH. The fitting of $N_m(T)$ and the parameters of $n_m(T)$ are shown in Fig. S1 and Table S4.**

### 3.3. Obtaining the differential freezing spectrum $n_m(T)$ from the experimental fraction of ice $f_{ice}(T)$ of insoluble ice nucleators using the HUB-backward code

**Sections 3.1** and **3.2** discuss how to obtain the differential spectrum from a target cumulative one. However, there are many cases where the results are presented as fraction of frozen droplets as a function of temperature, $f_{ice}(T)$. In these cases, the HUB-backward code can be used to obtain the optimized differential freezing spectrum $n_m^{optimized}(T)$ directly from $f_{ice}^{target}(T)$. **Supp. Section S5** illustrates this approach for the analysis of droplet freezing data for a sample of lignin (Bogler and Borduas-Dedekind, 2020) in which the IN participate in aggregation equilibria. Here, we exemplify the optimization the differential spectrum of cholesterol from experimental freezing data obtained at two cooling rates with droplets sampled from a single dilution.

The triangles and squares in **Fig. 11A** display the experimental $f_{ice}(T)$ obtained by sampling the freezing of hundreds of $120\mu L$ droplets pipetted from a suspension of cholesterol monohydrate crystals in contact with Teflon walls cooled at 0.18 K/min (triangles) and 0.06 K/min (squares) (Zhang and Maeda, 2022). The tripling of the cooling rate has a significant effect on the freezing of the droplets. In the analysis of drop-freezing experiments, it is assumed that each IN has a singular freezing temperature, independent of the cooling rate. However, ice nucleation is a stochastic process, and the

520 underlying distribution of freezing temperatures $P_u(T)$ strictly depends on both temperature and cooling rate, as slower rates give more time for the system to cross the nucleation barrier at warmer temperatures.

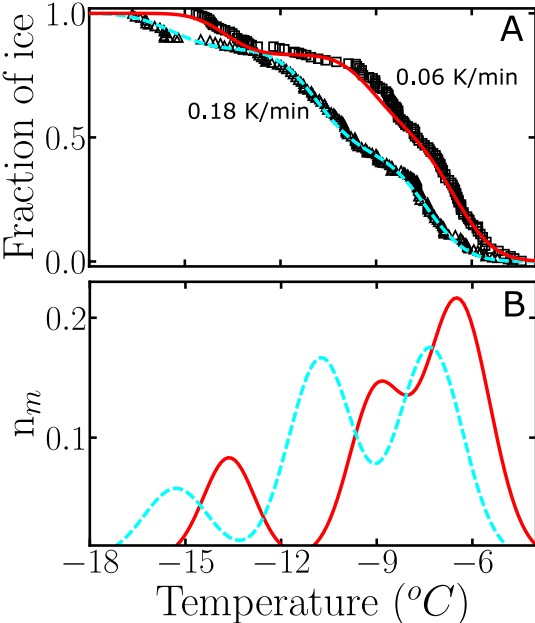

**Figure 11: Use of HUB-backward code to estimate the optimized differential freezing spectra $n_m^{optimized}(T)$ based on the fraction**
**of frozen droplets $f_{ice}^{target}(T)$ of cholesterol (Zhang and Maeda, 2022) at different cooling rates. Black circles and squares are experimental data, and cyan dashed and solid red lines represent the fit given by the HUB-backward code. The parameters of $n_m(T)$ are shown in Table S5.**

Our analysis of the freezing data of cholesterol monohydrate shows that even a three-fold change in the cooling rate can have significant impact on the differential spectrum (**Fig. 11B**). As expected, the modes of the three populations move
towards warmer temperatures upon decreasing the cooling rate. We note, however, that the shift of the peaks is not uniform; the middle one seems to be more sensitive to the cooling rate. Different sensitivity of the freezing rate of subpopulations to temperature has been also reported in simulations of nucleation data of minerals using the stochastic and modified singular frameworks (Herbert et al., 2014; Murray et al., 2011). The modified singular model proposes an empirical correction the relation between $f_{ice}(T)$ and $N_m(T)$ to account for the effect of the cooling rate on the shift of these quantities (Vali, 1994).
That analysis could be extended to the analysis of the subpopulations of IN obtained with HUB-backward. Moreover, it would be interesting in future studies to use the rate dependence of the mode of the subpopulations to extract the steepness of the nucleation barrier with temperature using nucleation theory (Budke and Koop, 2015), and to investigate the relationship between the cooling rate dependence of the differential spectrum obtained in the singular approximation with the interpretation of the same data modelled with the stochastic framework, such as in (Wright et al., 2013; Herbert et al., 2014).

 **4 Conclusions**

In this study, we present the HUB method and associated Python codes that model (HUB-forward code) and interpret (HUB-backward code) the results of droplet freezing experiments under the assumptions that each ice nucleating site in the sample has a characteristic nucleation temperature that is time-independent. The use of the singular approximation is the same as used by Vali (Vali, 1971; Vali, 2014, 2019) in his derivation of the ice nucleation spectra from data of fraction of frozen droplets. Different to previous implementations of the singular model, HUB accounts for the distribution of the number of IN in droplets at a given concentration, and uses extreme-value statistics to represent the effect of dilutions in the frozen fraction and freezing spectra. Our method and codes allow users to obtain an analytical differential freezing spectrum $n_m(T)$ from the experimental distribution of freezing temperatures, and vice versa. The differential freezing spectrum $n_m(T)$ is an approximant to the underlying distribution of ice nucleating temperatures $P_u(T)$, which provides a hub to connect the experimental freezing temperatures with interpretative physical analyses using kinetic models or nucleation theory that can be used to elucidate the mechanisms of nucleation and origins of these distributions.

HUB-forward predicts the cumulative ice nucleation spectrum $N_m(T)$ and fractions of frozen droplets $f_{ice}(T)$ from a known (or assumed) underlying distribution $P_u(T)$ of nucleation temperatures for the IN in the sample. The HUB-forward code can be used to investigate the effect of the number of droplets and dilutions on the temperature range of the cumulative freezing spectrum $N_m(T)$. Our analysis shows that the differential freezing spectrum $n_m(T)$ is identical to the underlying distribution of heterogeneous ice nucleation temperatures $P_u(T)$ only when sampling is complete. Measuring fewer droplets or fewer dilutions can result in a biased representation of the differential and cumulative spectra. HUB-forward predicts $f_{ice}(T)$ and $N_m(T)$ from a proposed distribution of IN temperatures, allowing its users to test hypotheses regarding the role of subpopulations of nuclei in the freezing spectra and providing a guide for a more efficient collection of freezing data.

HUB-backward uses a non-linear optimization method to find the differential freezing spectrum $n_m(T)$ that best represents the experimental target cumulative freezing spectrum $N_m(T)$ or fraction of frozen droplets $f_{ice}(T)$ in the experiments. The analytical form of the differential freezing spectrum $n_m(T)$ obtained from HUB-backward offers an interpretable physical basis. The interpretability of the results in terms of subpopulations provides an advantage over polynomial fitting and differentiation of $N_m(T)$. Indeed, we show that the HUB-backward code can be used to reveal and characterize the underlying number of IN subpopulations of complex biological samples (Snomax®, fungi *Fusarium acuminatum*, and birch pollen) and quantify the dependence of their subpopulations on environmental variables. Interestingly, our analysis evinces subpopulations that are not obvious to the eye and have not previously been identified in these samples. The robustness of the signals that correspond to these populations and their physical nature require further investigation.

We illustrate the use of HUB-backward to obtain the differential freezing spectrum $n_m(T)$ from the fraction of frozen droplets $f_{ice}(T)$ collected at a single concentration. We apply that analysis to demonstrate that $n_m(T)$ depends on the cooling rate. The shift of the peaks of the subpopulations to higher temperatures upon decreasing the cooling rate is not

unexpected, as longer waiting times allow for the surmount of the same nucleation barrier at warmer temperatures. By providing the temperature dependence of the mode spread and weight of the subpopulation peaks, HUB-backward can be combined with nucleation theory and other theoretical analyses to extract the steepness, and may even the distribution, of nucleation barriers that control the freezing process.

### Code availability

All codes, a user manual, and input files used in this project can be accessed at https://github.com/Molinero-Group/underlying-distribution

### Data availability

All data used in this project can be accessed by request to the authors.

### Author contribution

V. M., I. de A. R. and K. M. designed the project. I. de A. R. developed the model code and performed the simulations. I. de A. R. and V.M. prepared the manuscript with contributions from K.M.

### Competing interests

The authors declare that they have no conflict of interest.

### Acknowledgements

I. de A. R. and V. M. gratefully acknowledge support by AFOSR through MURI Award No. FA9550-20-1-0351. K. M. acknowledges support by the National Science Foundation under Grant No. (NSF 2116528) and from the Institutional Development Awards (IDeA) from the National Institute of General Medical Sciences of the National Institutes of Health under Grants #P20GM103408, P20GM109095.

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
