# Peer review of "HUB: A method to model and extract the distribution of ice nucleation temperatures from drop-freezing experiments"

_EGUsphere, 2022_

## Author Comment (AC1)

**Referee #1**:

We are very grateful to Dr. Borduas-Dedekind for her careful reading of the manuscript and for providing useful comments and suggestions that we have used to improve the manuscript. Below we give replies to all her comments and describe the modifications implemented in the revision (highlighted in red in this response and in the annotated manuscript for review).

Reviewer: General comments:

The authors present open access Python code to estimate the subpopulations of potential ice nucleating substances from data obtained by drop freezing assays. They present codes that have the potential to be quite important in further discussing the ice-nucleating ability of ambient samples from mineral dust to organic aerosols. I command the others for this important detailed work and for their clear writing. I'd like to raise a few discussion points and point out a few minor issues to be addressed prior to publication.

I'd first like to highlight what I thought were the most important contributions within this paper.

1. Clearly articulated problem to be addressed when using frozen fraction data (for example lines 15-16, 57-58, 75-80)

2. The dilution discussion (section 3.1) is particularly valuable, and the authors can make specific recommendations for the community to move forward in their data analysis.

3. The use of the HUB-forward code to estimate the presence of subpopulations.

Authors: We thank the reviewer for the positive assessment on our work and its presentation.

Reviewer: Here are my recommendations for improvement:

I struggled a little with the chosen terminology of the code. Why use the term "HUB"? What does the "underlying-based" mean in atmospheric science and/or in statistics? The forward/backward terminology was also not intuitive to me, and it's not clear why these terms pointing to a direction were used. Could there be better terms to be used such as "subpopulation determination" for HUB-forward? For example, the term could focus on the outcome of the code?

Authors: We chose the name "HUB" for the method because is short and points to the purpose of the code, which is to connect experimental data with theoretical interpretations. The "underlying-based" words in the "HUB" acronym refer to the outcome of the code, which is the underlying distribution of heterogeneous freezing temperatures. We coined this "underlying distribution" term to distinguish it from the differential spectrum, which approximates it only when the sampling is complete. We now strive to make the notation clearer in the manuscript.

Regarding the terminology "forward" and "backward", we understand that this may not be intuitive but hopefully is well understood after reading the manuscript and/or the manual for the code. In our code, "forward" refers to the determination of the fraction of frozen droplets and the cumulative freezing spectrum from the underlying distribution of heterogeneous freezing temperatures. "Backward" refers to the determination of the differential spectrum, the approximant of the underlying distribution of heterogeneous freezing temperatures, from experimental data.

Reviewer: I'd like to challenge an assumption made in the manuscript (for example on lines 138-139) about the role of dilutions. I think the presented data analysis method is best applied to ice nucleating substances that are intact. For example, mineral dust and P. Syringae proteins. However, there is literature on organic matter and dilution series where dilutions can potentially change the shape, form and composition of ice-nucleating sites. For example, (Bogler and Borduas-Dedekind, 2020) showed that dilutions of the macromolecule lignin influence the mass-normalized ice nucleating ability of the material. I would recommend that the authors expand on the idea that this dilution method is for intact ice-nucleating ability. Alternatively, the authors could also use the open access lignin data and see how their code performs (that would be cool actually!).

Authors: We concur with the reviewer and appreciate her suggestion. We now clarify this in the introduction of the method, page 6, lines 162-167:

"In our model, we assume that the underlying distribution of ice nucleating temperatures $P_u(T)$ does not change with the concentration of INs. This last condition is violated when IN are involved in chemical, aggregation, or solubility equilibria that alter the proportionality between their concentration and the dilution factor of the sample, resulting in a lack of overlap of the pieces of the cumulative spectra $N_m(T)$ obtained from different dilutions (Bogler and Borduas-Dedekind, 2020)."

We refer again to that the lignin data to section 3.3, lines 491-492:

"Supp. Section S5 illustrates this approach for the analysis of droplet freezing data for a sample of lignin (Bogler and Borduas-Dedekind, 2020) in which the IN participate in aggregation equilibria."

We add a new section S5 to the supporting section, which presents an analysis of different concentrations of the lignin paper, The analysis suggests that all concentrations correspond to the same IN, in concentrations that are not proportional to the dilution factor, supporting the interpretation of Bogler and Borduas-Dedekind 2020 of an aggregation equilibria that sequesters active IN as the amount of lignin material in solution is increased. We copy here the text and figures added to the SI:

**"S5 Modeling the differential spectrum of systems with chemical or phase equilibria using the HUB-backward code**

In the derivation of the HUB method, we have assumed that the average number l of IN per droplet is proportional to the dilution of the sample. The IN, however, can be involved in chemical or phase equilibria that would impact the proportionality between l and dilution. The result is a mismatch between the actual concentration of IN in solution and the total mass concentration of the sample. Ice nucleation of Lignin provides such an example (Bogler and Borduas-Dedekind, 2020). This data set combines two challenges. First, that the nucleation from the background water used to prepare the samples produces fraction of ice signal that is highly overlapped with that of the samples themselves (**Fig. S2-A**). This is a characteristic shared by all poor ice nucleants. Second, that the processing of the fraction of ice curves using Vali's equation (**Eq. 1.a** of the manuscript) result in cumulative spectra that do not overlap (**Fig. S2-B**). This implies that the concentration of IN is not proportional to the dilution (in this case, given by the total mass of organic carbon in the sample). In this example, the $N_m(T)$ curves seem to be parallel, suggesting that the nature of the IN is preserved across concentrations, but there is an aggregation equilibrium that makes its concentration increase in a sublinear manner with sample concentration. **Fig. S2-C** shows the differential spectrum $n_m$ obtained with HUB-backward from the fraction of frozen droplets of each concentration; these fits are shown with colored lines in **Fig. S2-A** and faithfully represent each of the $f_{ice}(T)$ curves. As expected from extreme value statistics with incomplete sampling,

the $n_m$(T) depends on the concentration (see also **Fig 2C** of the main text). The way the peaks move seems to be consistent with the analysis of extreme value statistics, but it is necessary to remove the background in order to do a better analysis.

[Figure]

**Figure S3: A) Fraction of ice for different concentrations of Lignin (Bogler and Borduas-Dedekind, 2020). Continuous lines represent the fitting of the fraction of frozen droplets obtained with the HUB-backward code using two subpopulations and Gaussian distributions as working basis. B) Cumulative freezing spectrum $N_m$ obtained from Ref. (Bogler and Borduas-Dedekind, 2020). C) The differential freezing spectrum $n_m$ obtained from the fitting shown by continuous lines in A). D) Points represent scaled cumulative freezing spectrum, and magenta dashed line is the fitting of the cumulative spectrum using HUB-backward.**

[Figure]

**Figure S4: Fitting of Nm using HUB-backward with two subpopulations with a left-tail Gumbel (left), and log-normal (right).**

**Table S6: Mean relative error (MRE) and parameters of the differential freezing spectrum obtained using the HUB-backward code from lignin data at various concentrations (Bogler and Borduas-Dedekind, 2020).**

|  | MSE | $T_{mode,1}$(°C) | $s_1$ | $T_{mode,2}$(°C) | $s_2$ | $c_2$ |
|---|---|---|---|---|---|---|
| 2 mg | $2\times10^{-4}$ | -21.05 | 2.68 | -23.33 | 1.20 | 0.49 |
| 20 mg | $2\times10^{-4}$ | -21.46 | 1.20 | -19.24 | 2.85 | 0.48 |
| 200 mg | $2\times10^{-4}$ | -19.84 | 1.11 | -17.51 | 2.09 | 0.65 |
| background | $2\times10^{-4}$ | -24.55 | 0.70 | -22.40 | 2.95 | 0.59 |
| $N_m$ fitting (Gaussian) | 0.010 | -22.48 | 0.74 | -18.93 | 2.10 | 0.14 |
| $N_m$ fitting (log-normal) | 0.015 | -24.64 | 0.79 | -19.42 | 0.77 | 0.05 |

Reviewer: I also wonder about the choice of Gaussian distributions (Eq3) for the freezing temperatures of populations of IN. Why not log normal? Lines 123-124 mention that other types of normalized distributions could be used, so it would be important to justify this choice. From my own understanding, ambient samples/datasets are typically log normal. See also (Andersson, 2021).

Authors: We selected the Gaussian distribution (or normal distribution) because it is a widely used and well-understood distribution that describes many real-world phenomena. Additionally, it has desirable mathematical properties, such as being symmetric and having a defined mean and variance, which can simplify the analysis of the data. Additionally, we had previously tested the data with left-tailed Gumbel distribution and found it inadequate to represent the differential spectrum. Both Gumbel with left tail and log-normal are asymmetric distributions, with the tail pointing in opposite directions. These distributions have a lower decay towards lower temperatures and higher temperatures, respectively.

Following the suggestion of the reviewer we now add the (right-tailed) log-normal, left-tailed Gumbel distributions to the HUB-forward and HUB-backward codes. We add the following additional text to the manuscript, after introducing the normal (Gaussian) form of the populations, page 6, lines 160-163

"We also provide in the HUB code the option for the user to use the log-normal distribution, which has a tail towards higher temperatures, or the left-tailed Gumbel distribution, which has a tail towards lower temperatures. In our model, we assume that the underlying distribution of ice nucleating temperatures $P_u(T)$ does not change with the concentration of INs."

We updated the code in GitHub with these new options, that are offered to the user in the interactive version of the code.

We further tested the log-normal as a basis in the HUB-backward code to fit the fungi and bacteria data (magenta line below shows the analysis with log-normal). In these examples, the log-normal distribution does not provide a better fit to the data. Likewise, the reviewer can see in the SI introduced in response to the previous question that the log-normal distribution performs slightly worse than the Gaussian in representing the data.

[Figure]

Reviewer: The manuscript is well written and well-motivated. The flow could be improved with more subsections to be able to find the information rapidly for the future reader. For instance, after reading the paragraph at lines 178-186 – I would have been interested to see this code applied in the following section. There could also be a Method section for the details of the math and then a Results and Discussion section with subsections for categories related to recommendations like dilutions series, subpopulations, etc. Subsections within pages 9-10-11 would also help the flow.

Authors: We hope that the last paragraph of the introduction, which details the contents of the sections, serves as the guide that the reviewer is asking for. We now clarify there that Section 2 is the methods section. That section provides already all the mathematical formulations used in the paper. We have included the methods section 2 –examples to help readers better understand the methodology, which we believe is a more effective pedagogical approach to present the algorithms. Section 3 is the results section, where we applied HUB-forward and HUB-backward to address multiple scenarios that illustrate their power and potential.

Reviewer: There are additional references that I would encourage the authors to consider, and I've added them throughout my specific comments below.

Authors: Thank you for making us aware of these studies.

**Specific comments:**

Reviewer: Title: The title might be improved by specifying the types of ice nuclei as well as either defining HUB or removing the acronym.

Authors: We prefer to keep the title as initially proposed: the name of the method (HUB), and then the description of what it does ("a method to model and extract the distribution of ice nucleating temperatures from drop freezing experiments"). Keeping HUB in the title makes it easier for the readers to identify the manuscript with the code, which we hope will be of interest to many in the ACP community. The

BINARY (Budke and Koop, 2015) and CHILL+ (Nguyen and Molinero, 2015) are well-known examples of using backronyms in the title of a paper to identify the code or technique.

Reviewer: Lines 32-36 has a rather random assortment of references of some drop freezing assays. I can refer the authors to a < 2021 comprehensive table of reported techniques: Table 1 in (Miller et al., 2021)

Authors: We thank the reviewer for pointing out the comprehensive list of reported techniques in the literature. We replace the long list by (page 2, line 34):

"A comprehensive report of various drop freezing techniques can be found in (Miller et al., 2021)."

Reviewer: Line 36: I would also comment that many drop freezing techniques are also used for ambient measurements with unknown concentrations and unknown surface area like sea surface samples and ambient aerosols. How would the authors use their code on these types of samples?

Authors: Our code (and Vali's formulation, in general) can be used even when the absolute concentrations or areas of the IN are not known, provided that the user knows the relative concentration of the dilution series of the parent sample. The foundation of this is apparent in eq. 1a, where the normalizing factor X could have any arbitrary units (e.g. some pollen data is presented in units of grains of pollen, and then the cumulative spectrum would be in units of IN per grain of pollen). We now clarify this in the introduction, lines 70-75:

"For soluble INs, the normalization factor is commonly defined by the mass of the ice nucleating material $X = \rho \, (V_{drop}/d)$, where $\rho$ is the density of the initial solution, $V_{drop}$ is the droplet volume and $d$ is the dilution factor (Kunert et al., 2018). The IN surface area per drop, $X = A_{drop}$, is sometimes used as normalization factor for insoluble INs (e.g., dust, crystals), although it is challenging to measure the total IN surface area accurately (Knopf et al., 2020). We note that Eq. 1a can be used even when the absolute concentrations or areas of the IN are unknown, provided that the user knows the relative concentration of the dilution series derived from a parent sample."

Reviewer: Lines 42-46 discuss the role of cooling rate which is important in data evaluation. I would encourage the authors to comment and reference (Wright et al., 2013). Also relevant to the discussion on lines 440-447.

Authors: We now add to the introduction a discussion of stochastic modeling of ice nucleation data that references the work of Wright and Petters and others, and also extend the discussion of section 3.3 to highlight future research opportunities:

"Our analysis of the freezing data of cholesterol monohydrate shows that even a three-fold change in the cooling rate can have significant impact on the differential spectrum (**Fig. 11B**). As expected, the modes of the three populations move towards warmer temperatures upon decreasing the cooling rate. We note, however, that the shift of the peaks is not uniform; the middle one seems to be more sensitive to the cooling rate. Different sensitivity of the freezing rate of subpopulations to temperature has been also reported in simulations of nucleation data of minerals using the stochastic and modified singular frameworks (Herbert et al., 2014; Murray et al., 2011) The modified singular model proposes an empirical correction the relation between $f_{ice}(T)$ and $N_m(T)$ to account for the effect of the cooling rate on the shift of these quantities (Vali, 1994). That analysis could be extended to the analysis of the subpopulations of IN obtained with HUB-backward. Moreover, it would be interesting in future studies to use the rate dependence of the mode of the subpopulations to extract the steepness of the nucleation barrier with temperature using nucleation theory (Budke and Koop, 2015), and to investigate the

relationship between the cooling rate dependence of the differential spectrum obtained in the singular approximation with the interpretation of the same data modelled with the stochastic framework, such as in (Wright et al., 2013; Herbert et al., 2014)."

Reviewer: Eq1b and differential freezing spectra have been discussed previously in (Creamean et al., 2019) and so this reference should be added and discussed."

Authors: Eq1b is the formulation presented by Vali in 1971 (we now add the citation). We now mention Creamean et al., 2019 in the paragraph that discusses the differential spectra in terms of populations, lines 110-113:

"While several studies have broadly defined populations by the range of nucleation temperatures they encompass(Turner et al., 1990; Creamean et al., 2019) or the origin of the sample (Steinke et al., 2020), there is currently no simple procedure to identify and quantify subpopulations or classes from cumulative freezing spectra $N_m(T)$."

Reviewer: Scheme 1: "I_u" is not defined. I also think this scheme could be improved by using graphics instead of terms. In other words, the authors could show a frozen fraction graph and show the type of graphs that may be generated based on their code. (especially since different research groups use different terms, a graphical visualization would be helpful – and could also serve as a TOC graphic)

Authors: Thanks for pointing this out, the definition of "I_u" appeared only later in section 2.2. We now refer it for the first time in caption of Figure 1, and point to section 2.2. where the equation is derived:

**"The intrinsic cumulative spectrum $I_u(T)$ is proportional to $\int_{T_m}^{T} P_u(T')dT'$ (section 2.2)."**

We also follow the suggestion of the reviewer and update Figure 1 with figures that illustrate the type of generated graphs from our analysis:

[Figure]

Reviewer: Lines 102-109 could be omitted entirely as these sentences are redundant (more appropriate for a thesis rather than a manuscript)

Authors: We prefer to keep the paragraph about the organization of the manuscript, to allow readers to easily find where the methods (section 2) and applications (section 3) are discussed in this study and access them according to their interests and needs.

Reviewer: The idea of Eq2 and the sum of all parts has been nicely discussed in (Steinke et al., 2020) and the authors should consider mentioning this work.

Authors: We now mention (Steinke et al., 2020) in the paragraph about populations cited above.

Reviewer: Figure 1 – PMF should also be defined in the text. It's also difficult to see the black line in figure 1. Perhaps making it bold would help?

Authors: We update Fig. 2 B-C with a magenta dashed line to enhance the contrast and add the following statement to the manuscript:

"shows the probability mass function (PMF)"

Reviewer: Would it be worth relegating the tables to the SI? Some of the values could be added directly onto the graphs for instance.

Authors: There are only two tables in the manuscript (six more are already in the Supporting Information). These tables provide the uncertainty in the fit of the cumulative spectra and important information of the populations that is relevant to the main arguments of the paper, that should be readily accessible to the reader.

Reviewer: Line 330-331 – there is much value in having code now to support this claim! Well done to the authors.

Authors: Thank you, we appreciate and share your enthusiasm for highlighting the importance of dilutions in the calculation of the freezing spectra.

Reviewer: Lines 335-343 – excellent recommendations

Authors: Thank you!

Reviewer: Figure 6 – specify in the caption the difference between panels A, B, C and D.

Authors: We thank the reviewer for pointing that out. To clarify this, we add the following statement to the caption of Fig. 6:

"Panels A, B, C and D were computed with a different number of dilutions."

Reviewer: Line 368-369: it would be worth describing how the choice of "2 subpopulations" was made. If I understood correctly, it was previously optimized? Or are the authors sourcing this information another way? It would be worth clarifying.

Authors: The choice of "2 subpopulations" was not sourced from any previous optimization but was rather determined through our analysis of the data. In our study, we performed a series of tests to determine the best fit for the data and found that a two-subpopulation model provided the best explanation of the observations. This conclusion was drawn based on a number of criteria, including the mean squared error, the goodness of the fit, and the simplicity of the model. Table 2 shows the mean squared error of each model.

We now add an extra column to Table 2 that indicates explicitly the number of subpopulations for each optimization.

Reviewer: Figure 8 – there's an error on the panel labels in the caption. ABC should be ACD.

Authors: Thank you for bringing this to our attention. We have reviewed the caption of Figure 8 and could not find the error you have described. However, we now add the panel letters and not just left and right, to the caption of figures 7 and 8 to avoid misinterpretations.

Reviewer: Line 386-387 – why were some points omitted from the optimization procedure?

Authors: The differential spectrum obtained from the sparsely sampled black data of the original submission and the total (black plus gray) data points are almost identical. We now indicate this in the manuscript, lines 464-466:

"Section S4 shows that the differential spectrum optimized from the whole data set and its sparse sampling are almost identical, because HUB-forward interpolates and smooths the input data to produce an equispaced data set.

We here copy the new section S4:

**"S4 Effect of sparse sampling of a cumulative spectrum on the estimation of the differential spectrum with HUB-backward**

**Figure 9E-F** presents an analysis of a sample of the pollen data of (Dreischmeier, 2019). **Supp. Fig. S2** shows that the analysis of the full data set shown **Fig. 9E** produces an almost identical differential spectrum, because HUB-backward interpolates the input data to produce a smooth and equally spaced data set (**Supp. Fig. S3**).

[Figure]

**Figure S2: Effect of sparsely sampling a dense data set. A) cumulative spectra (black circles) of pollen from (Dreischmeier, 2019) and its fitting to two populations with HUB-backward fitting (magenta line). B) differential spectrum derived by that analysis from the analysis of all experimental data points (magenta line) is almost indistinguishable from the one obtained by sparsely sampling the data set (blue line, also shown in Fig 9F). C) cumulative spectra (black circles) of pollen from (Dreischmeier, 2019) used as input for HUB-backward and interpolated data using the default parameters of the code (magenta line)."**

Reviewer: Figure 9 Panel A is arguably an important graph and would benefit from being highlighted separately (perhaps moving the other panels to the SI?).

Authors: We find all three examples in figure 9 important, as they illustrate different aspects of the optimization (number and relative weights of populations, noisy data…). We appreciate your recognition of the importance of the data for fungi, which is our focus in a separate study, to which we now refer in lines 426-427:

"We refer the reader to (Schwidetzky et al., 2023) for an interpretation of the size of the ice nucleating surface of *F. acuminatum* based on its differential spectrum and nucleation theory."

**Reviewer: References:**

Andersson, A.: Mechanisms for log normal concentration distributions in the environment, Sci Rep, 11, 16418, https://doi.org/10.1038/s41598-021-96010-6, 2021.

Bogler, S. and Borduas-Dedekind, N.: Lignin's ability to nucleate ice via immersion freezing and its stability towards physicochemical treatments and atmospheric processing, Atmospheric Chemistry and Physics, 20, 14509–14522, https://doi.org/10.5194/acp-20-14509-2020, 2020.

Creamean, J. M., Mignani, C., Bukowiecki, N., and Conen, F.: Using freezing spectra characteristics to identify ice-nucleating particle populations during the winter in the Alps, Atmospheric Chemistry and Physics, 19, 8123–8140, https://doi.org/10.5194/acp-19-8123-2019, 2019.

Miller, A. J., Brennan, K. P., Mignani, C., Wieder, J., David, R. O., and Borduas-Dedekind, N.: Development of the drop Freezing Ice Nuclei Counter (FINC), intercomparison of droplet freezing techniques, and use of soluble lignin as an atmospheric ice nucleation standard, Atmospheric Measurement Techniques, 14, 3131–3151, https://doi.org/10.5194/amt-14-3131-2021, 2021.

(Steinke et al., 2020), I., Hiranuma, N., Funk, R., Höhler, K., Tüllmann, N., Umo, N. S., Weidler, P. G., Möhler, O., and Leisner, T.: Complex plant-derived organic aerosol as ice-nucleating particles – more than the sums of their parts?, Atmospheric Chemistry and Physics, 20, 11387–11397, https://doi.org/10.5194/acp-20-11387-2020, 2020.

Wright, T. P., Petters, M. D., Hader, J. D., Morton, T., and Holder, A. L.: Minimal cooling rate dependence of ice nuclei activity in the immersion mode, Journal of Geophysical Research: Atmospheres, 118, 10,535-10,543, https://doi.org/10.1002/jgrd.50810, 2013.

**Citation**: https://doi.org/10.5194/egusphere-2022-1242-RC1

---

## Author Comment (AC2)

**Referee #2**:

We thank the anonymous reviewer for their useful comments and suggestions that we have used to improve the manuscript. Below, we give replies to all comments and describe the modifications implemented in the revision (highlighted in red in this response and the annotated manuscript for review).

Reviewer: The manuscript, "HUB: A method to model and extract the distribution of ice nucleation temperatures from drop-freezing experiments", presents a way to simulate droplet freezing to calculate frozen fractios and ice active site density. As stated by the authors, their main goal is to link data to theory using 1 or more probability distributions. Also, they aim to described how to sufficiently sample the ice nucleation spectrum, which is interpreted as using a certain number of droplets in experiments and performing dilutions series. This will result in not too much noise in the calculated cumulative or differential ice nucleation spectra. They analyze previous data to show that a distribution of freezing events can change when solution pH changes and when cooling rate changes.

Authors: We thank the reviewer for the summary. The main goal in this paper is to provide a methodology to generate cumulative freezing spectra and fraction of ice distributions from a proposed population model, and vice versa. We want to make clear here that the theoretical interpretation of these differential spectra (and their fully sampled underlying distribution of nucleation temperatures) is not a goal of the present study. We edited the wording of the goals of this study to make this clearer to the readers, lines 100-103 and 114-116:

"The first goal of the present study is to provide a strategy to optimize the sampling of drop-freezing experiments to derive interpretable differential spectra that is a good approximant of the underlying distribution of heterogeneous ice nucleation temperatures of the sample."

"The second aim of our study is to map the cumulative freezing spectrum $N_m(T)$ into the differential spectrum $n_m(T)$, in terms of subpopulations that may correspond to different physical nucleation sites in the sample."

Reviewer: Unfortunately, I see minimal merit for publishing this study and cannot recommend publication, unless significant revision is made. Perhaps a complete resubmission should be done. A Monte Carlo simulation to predict frozen fraction and $n_s$ or $n_m$ is not new and their main goals have already been accomplished by other work (Vali, 1971; Wright and Petters, 2013; Knopf and Alpert, 2013; Herbert et al., 2014; Vali, 2019; Fahy et al., 2022a; Fahy et al., 2022b). By no means is this list of references complete, the authors can look up their cited references and other studies that cite these to find numerous other models.

Authors: We appreciate that the use of Monte Carlo simulations to predict and interpret frozen fractions has been explored in previous studies, such as some of those provided by the reviewer.

We extensively revise the introduction of the revised manuscript to provide a brief account of other approaches. Lines 42-61:

"Historically, there have been two interpretations of the dispersion of nucleation temperatures in heterogeneous freezing experiments. The first approach suggests that the stochastic nature of the nucleation process dominates the variability in freezing temperatures (Bigg, 1953; Carte, 1956), while the second approach assumes that the dispersion in temperatures mostly arises from a distribution of nucleation sites (Fletcher, 1969), each with a deterministic, singular nucleation temperature (Levine, 1950; Vali and Stansbury, 1966). Variability in the temperature, volume, and amount of ice-nucleating

particles per droplet can also contribute to the dispersion of freezing temperatures (Vali, 2019; Knopf et al., 2020). There is consensus now that both stochastic effects and sample heterogeneities contribute to the distribution of freezing temperatures, and both approaches are used for the modelling of drop-freezing experiments (Vali, 1971; Marcolli et al., 2007; Niedermeier et al., 2011; Murray et al., 2011; Broadley et al., 2012; Wright and Petters, 2013; Herbert et al., 2014; Harrison et al., 2016; Alpert and Knopf, 2016; Vali, 2019; Fahy et al., 2022a). Stochastic modelling of the freezing curves is based on predicting the survival probability of liquid water containing IN as a function of supercooling, and requires a model for the temperature dependence of the nucleation rate of the IN components. These models have been solved numerically or evolved with Monte Carlo simulations to interpret or resolve the distribution of ice nucleation properties of minerals (Marcolli et al., 2007; Murray et al., 2011; Broadley et al., 2012; Wright and Petters, 2013; Herbert et al., 2014; Harrison et al., 2016) and organics (Zobrist et al., 2007; Alpert and Knopf, 2016) and to perform parametric bootstrapping of experimental data (Wright and Petters, 2013; Harrison et al., 2016). The advantage of the stochastic modelling approach is that it enables a direct link to microscopic properties of the nuclei and can account for the cooling rate dependence of the $f_{ice}(T)$ data. However, the requirement of a model for the freezing rates and their distribution across the sample hinder their interpretability and accuracy at reproducing the experimental freezing curves, particularly in complex samples containing multiple populations."

We note that HUB-backwards does not use Monte Carlo sampling, but a numerical optimization procedure. HUB-forward approach is based on Monte Carlo as we are randomly sampling the freezing temperatures from a distribution. However, our method is fundamentally different to those in previous studies. On one hand, the HUB method does not rely on nucleation theory nor does it assume a model for the nucleation rates. The HUB method is based on Vali's 1971 formulation of singular freezing, but it expands on Vali's modeling approach by introducing the random sampling of IN in the droplets, the use of analytical functions for IN subpopulations and –most critically- the use of extreme-value statistics. The latter is paramount to consider the effect of dilutions on the freezing spectra and derive a relation between the underlying distribution of nucleation temperatures (or its approximant, the differential spectrum) and the cumulative spectrum and fraction of droplets crystallized as a function of temperature.

To our knowledge, the fraction of frozen droplets and the cumulative spectra based on extreme value sampling have not been explored in any previous study. Our approach provides a unique perspective on the relationship between the underlying distribution of freezing temperatures, the fraction of frozen droplets, and the cumulative freezing spectra. Our method illustrates how diluting changes the sampling, making it a tool for a better design of experimental protocols, and enable an efficient recovery of analytical expressions for the differential spectrum that can be interpreted using theories and/or microscopic or population models.

We now elaborate on other aspects of previous methods and note that –to our knowledge extreme-value statistics has not been previously used for the modeling of ice nucleation data, lines 77-93:

"The differential spectrum identifies the density of IN active at each temperature, and was identified by Vali as the central quantity that can be derived and interpreted from drop-freezing experiments (Vali, 1971; Vali, 2019).

The determination of the differential spectrum from the cumulative one by finite differentiation is subject to significant noise, requiring a careful selection of the temperature intervals and extensive sampling (Vali, 2019). As stochastic effects are not considered in the singular temperature formalism, the cumulative and differential spectra should –in principle- depend on the cooling rate (Vali, 1994). The stochastic nature of ice nucleation, combined with the uncertainties associated with the experimental measurements (e.g., different droplet volumes, inhomogeneous samples, different detection efficiencies),

can produce significant variations in the cumulative freezing spectra, that result in large uncertainties in $n_m(T)$. Parametric and nonparametric bootstrapping based on the singular approximation and Monte Carlo simulations have been used to estimate confidence intervals in freezing spectra measurements (Vali, 2019; Fahy et al., 2022a; Fahy et al., 2022b).

A central assumption of the singular freezing approximation is that the freezing of a droplet containing multiple INs is promoted by the IN with the highest nucleation temperature (Levine, 1950). This results in extreme-value statistics for the sampling of the nucleation temperature of the droplets (Sear, 2013). The extreme-value sampling is apparent in the concentration dependence of $f_{ice}(T)$ in experiments (Marcolli et al., 2007; Budke and Koop, 2015; Kunert et al., 2018; Lukas et al., 2022). However, to our knowledge, the impact of extreme-value statistics has not been considered in the singular modelling of drop-freezing experiments."

Reviewer: Furthermore, the authors provide no quantitative uncertainty and no error bars, confidence intervals or prediction bands of the simulated experiments, therefore, no assessment of accuracy in this work.

Authors: We agree with the reviewer on the importance of error bars. In the original submission we provided the mean square errors (MSE) used to evaluate the goodness of the fit. We now add error bars to the data presented in Table 1 and Table 2 to indicate the variability of the statistical estimates.

Reviewer: I did find simulating a dilution series and previous data with different pH interesting. The other new aspect is showing that the 3 probability distributions for cholesterol freezing is time dependent. For this paper to be acceptable, the authors should greatly expand their work. A resubmission should include a reproduction of other ice nucleation Monte Carlo models.

Authors: We are happy to hear that the reviewer found the simulation of the dilution series and the correlation of cholesterol freezing with pH interesting and appreciate the reviewer's recognition of the time-dependent nature of the three probability distributions for cholesterol freezing. We expand the discussion of section 3.3 to make a connection of the results of time dependence of cholesterol with other studies and highlight opportunities for future research, lines 512-524:

"Our analysis of the freezing data of cholesterol monohydrate shows that even a three-fold change in the cooling rate can have significant impact on the differential spectrum (**Fig. 11B**). As expected, the modes of the three populations move towards warmer temperatures upon decreasing the cooling rate. We note, however, that the shift of the peaks is not uniform; the middle one seems to be more sensitive to the cooling rate. Different sensitivity of the freezing rate of subpopulations to temperature has been also reported in simulations of nucleation data of minerals using the stochastic and modified singular frameworks (Herbert et al., 2014; Murray et al., 2011) The modified singular model proposes an empirical correction the relation between $f_{ice}(T)$ and $N_m(T)$ to account for the effect of the cooling rate on the shift of these quantities (Vali, 1994). That analysis could be extended to the analysis of the subpopulations of IN obtained with HUB-backward. Moreover, it would be interesting in future studies to use the rate dependence of the mode of the subpopulations to extract the steepness of the nucleation barrier with temperature using nucleation theory (Budke and Koop, 2015), and to investigate the relationship between the cooling rate dependence of the differential spectrum obtained in the singular approximation with the interpretation of the same data modelled with the stochastic framework, such as in (Wright et al., 2013; Herbert et al., 2014)."

Regarding the request by the reviewer that we "include a reproduction of other ice nucleation Monte Carlo models", we note that the HUB-backward optimization that recovers the differential spectrum is not based on Monte Carlo sampling. Only HUB-forward uses Monte Carlo, in the form of random sampling

of a proposed underlying distribution, and its use is for the study of the role of dilutions and number of droplets on the cumulative spectrum. We are not aware of other Monte Carlo studies that have performed an analysis of the role of dilutions on the prediction of the differential spectrum that we could compare with.

Reviewer: To relate data and theory, they should derive a mathematical link between their model and theory as this is only discussed in a few sentences in passing despite being a main goal.

Authors: The focus of this manuscript is not deriving a relation between the distribution of nucleation temperature and nucleation theory, but to present a method to extract the differential spectrum and assess the quality of the sampling (as indicated by the two explicit goals in the introduction section of the manuscript).

For a connection between HUB results and theory we now refer in the paper the reader to our recent study of ice nucleation by *F. acuminatum*, where we connect the differential spectrum obtained with HUB-backward analysis of droplet freezing experiments with the prediction of classical nucleation for finite surfaces, to predict the size of the ice nucleating complex responsible for the exceptional ice nucleation ability of this fungus, lines 426-427:

"We refer the reader to (Schwidetzky et al., 2023) for an interpretation of the size of the ice nucleating surface of *F. acuminatum* based on its differential spectrum and nucleation theory."

that manuscript includes also extensive physico-chemical characterization that allows us to determine the size of the ice nucleating protein (INP), which we combine with the predictions of the theory to derive the number of INP involved in the nucleating complex of *F. acuminatum*.

Reviewer: Finally, they should include an uncertainty analysis of both model and experimental results and provide new data to test their model. new data could be droplet freezing experiments and dilution series data where they know exactly what the subpopulations are before they start an experiment.

Authors: As it is not possible to know the subpopulations before starting an experiment, we use synthetic data to perform the validation and examples of the use of the model in section 2 and 3.1, and only use experimental data in sections 3.2 and 3.3 to illustrate the use of the methodology. All experimental results in this paper are taken from the literature, we are not providing new experimental data but rather a numerical methodology to analyze the data. Other methods in the literature, e.g. bootstrapping can be used to complement the analysis and derive error bands for the data that can be used to produce independent HUB-backward optimizations that would give a more complete account of the uncertainties in the estimations. HUB is not meant to replace other methods already available in the literature, but to be used synergistically with them.

Reviewer: Major Comments

1) l. 18 "first available code". This is not the first available code to predict frozen fraction or cumulative ice nucleation spectra from a probability distribution. In addition, many of other authors have made their code available, as it is a requirement for data and code availability in most journals and research grants. Therefore, this phrase or anything else similar must be removed from the manuscript.

More generally, it is likely the first time something has been done when a manuscript is published. Yes, how the authors define their probability distribution is unique, but it is distracting and unnecessary to say it is the "first time".

Authors: We understand the reviewers' point and remove all claims of first from the manuscript.. However, we would like to emphasize here that our code is the first to utilize Vali's approach and extreme value statistical analysis without making any physical assumptions about the system. This is a unique contribution and sets our code apart from other available codes. We have revised the manuscript to accurately make clearer the differences between our approach and previous ones.

Reviewer: l. 15-16 "no rigorous statistical analysis…to obtain a well-converged $n_m$ that represents the underlying distribution $P_u(T)$.". This is not true. Uncertainty are calculated by the previously mentioned studies as well, and with them one can know what is or is not in agreement, what is representative or converged or not. What the authors considers as well-converged is their opinion as there is no uncertainty estimation to claim agreement or not.

Authors: We acknowledge that previous studies have calculated uncertainties, but we would like to clarify that our statement refers to the lack of a rigorous statistical analysis to determine the number of dilutions necessary to adequately sample the entire freezing temperature spectrum. This information is crucial in order to obtain a well-converged estimate of the underlying distribution.

In our view, the issue is not the absence of uncertainty estimation, but rather the absence of a systematic approach to determine the number of dilutions required to achieve a representative sample.

Reviewer: 2) l. 13 "Underlying distribution" The word underlying has the meaning of something that is real or fundamental to nature. Defining probability distribution of different populations whether this is one, two or ten populations is not demonstrated here to be anything fundamental or real. "Underlying distribution" also has the meaning of something that is not immediately obvious. Whether there is one or more than one distribution (subpopulation) of freezing temperatures is always pre-defined by the authors. In other words, they authors no not derive the number of subpopulations, it is always prescribed for their forward and backward code. This is assumed not underlying.

Authors: "Underlying" means that is the one from which the freezing data is generated, and we hope that the revisions in the manuscript make this clearer.

The reviewer is correct: the subpopulations used for the analysis are assumed, and as such the differential spectrum is always an approximant of the underlying distribution. The distinction between the underlying distribution (that is known in the synthetic data sets we prepare to assess the method, but not in the case of real experimental data) and the differential spectrum is central to our study. Based on the comment by the reviewer, we have revised the abstract –where Pu is first introduced- to make this distinction clear:

"The differential freezing spectrum $n_m(T)$ is an approximant to the underlying distribution of heterogeneous ice nucleation temperatures $P_u(T)$ that represents the characteristic freezing temperatures of all IN in the sample."

Reviewer: 3) Units. I cannot understand the units in Eqn 5. I know the units of $n_m$ as Mass$^{-1}$, and the units of the differential spectrum $N_m$ as Mass$^{-1}$ Temperature$^{-1}$. In Eqn 5, the unit of Pmax then has to be Temperature$^{-1}$ for the frozen fraction to be dimensionless? Would the authors include an equation of Pmax in the manuscript, and check units throughout.

Authors: Yes, the reviewer is correct and we thank them for pointing out that we had not indicating the units. We now clarify the this in lines 156-157:

"The units of $P_u(T)$ are, same as for $n_m$(T), i.e. those of the cumulative spectrum divided by a unit of temperature, but are generally omitted in what follows."

And in lines 188-190:

"$P_{max}^\lambda(T)$ also have the same units as the differential spectrum $n_m$(T) similarly to $P_u(T)$, but we have chosen to omit the units for simplicity in our analysis."

Reviewer: 4) There are no uncertainty estimate in this manuscript.

Authors: We have included error bars for the modes, widths, and weights of the subpopulations in Table 1 and Table 2.

Reviewer: Minor Comments

1) It is common practice, that the cumulative spectra is a lower case n(T). When normalized to mass, it is n$_m$ and when normalized to surface area it is n$_s$. Please change this accordingly.

Authors: All the experimental data we use presents data per mass. We have seen various notations in the literature. We here use lower case n for the differential spectrum and upper case N for the cumulative. We now clarify in the introduction, lines 72-75, that the spectra normalized per area has a different name:

"The IN surface area per drop, $X = A_{drop}$, is sometimes used as normalization factor for insoluble INs (e.g., dust, crystals), resulting in a cumulative spectrum per area denoted as $N_s(T)$. However, it is challenging to measure the total IN surface area accurately (Knopf et al., 2020). We note that Eq. 1a can be used even when the absolute concentrations or areas of the IN are unknown, provided that the user knows the relative concentration of the dilution series derived from a parent sample."

Reviewer: 2) l. 55-60 How a probability distribution connects ice nucleation experiments and theory needs to be cited and derived. This statement is unsupported. The number of freezing events defines uncertainty, and how many droplets is or is not good enough is opinion without a rigorous definition.

Authors: The sentence highlighted by the reviewer "The underlying distribution $P_u(T)$ is akin to a hub that connects the experimental freezing temperatures to physical analysis based on nucleation theory or kinetic and equilibrium models that can elucidate the mechanisms and origins of the distributions of INs (**Fig. 1**)." is to motivate the importance of the underlying distribution, and extracting good approximants of it from well-sampled experiments. The analysis of the differential spectrum (as approximant for Pu) in terms of theories is outside the scope of this work. However, an integration of the results of HUB and nucleation theory can be found in (Schwidetzky et al., 2023).

Regarding your statement that the number of freezing events defines uncertainty and that the appropriate number of droplets is opinion without a rigorous definition, we respectfully disagree. The number of freezing events, which directly relates to the concentration of ice nucleating particles, and the optimal number of droplets are a fundamental metric in ice nucleation experiments, and our code explores that, and the results of the analysis are detailed in the paper.

Reviewer: 3) l. 70 "based on empirical bootstrapping" What was the most important in (Fahy et al., 2022a) is the non-parameteric bootstrapping was used, i.e. without any prior probability distributions

needed. Here, the authors need to assume a distribution (l. 121) and already puts in bias to their methods. They have to define the number of subpopulations (l. 171), again biasing their model.

Authors: We agree that non-parametric bootstrapping is a powerful statistical tool and appreciate your comment that it assumes the sample is representative of the population and that the observations are independent and identically distributed. However, we would like to clarify that our method is distinct from non-parametric bootstrapping, and that it addresses the issue of identically distributed data in a completely different way. Specifically, our method is based on the extreme value statistics approach, which allows us to model the fraction of frozen droplets data for a given concentration as a function of the tail of the underlying distribution.

We indeed expect the user to define the number of populations and make judgement on the quality of the result. The good agreement between the input cumulative spectra and the ones obtained from our simulations (same for the fraction of ice vs temperature), supports that the analysis with subpopulations is not only conceptually simple but also quite effective.

We would also like to note that the quality of results obtained through non-parametric bootstrapping may indeed depend on the size and distribution of the original sample, as well as the number of resamples generated, as discussed by Fahy et al.

Reviewer: 4) l. 77 The authors are not the first with a way to quantify subpopulations or different types of ice active sites or multi-component freezing to put it another way. There are too many studies to cite about mineral dust, pollen, bacteria, sea spray aerosol particles, washing water etc… A method to quantify subpopulations was done as early as 4 decades ago (Yankofsky et al., 1981).

Authors: Thank you for bringing up previous studies on quantifying subpopulations; we now add more references to some of these in the manuscript. Yet, our method fundamentally differs from previous methods in that it quantifies subpopulations in terms of sampling weights. Sampling weights are a statistical technique used to adjust for the fact that different members of a population may have different probabilities of being sampled. In our method, we use sampling weights to account for the fact that different ice active sites or multi-component freezing events may have different probabilities of being measured, due to differences in their size, composition, or other factors. This allows us to estimate the relative abundance of different subpopulations within a sample, and to make inferences about the overall population of ice active sites or multi-component freezing events.

The use of analytical functions for the populations also allows to determine the differential spectrum from the cumulative one without the need of binning or coarsening of the data. Vali 2019 presents an excellent discussion of the effect of coarsening on the calculation of the differential spectrum.

Reviewer: 5) l. 99 What is the difference between an underlying distribution and a true underlying distribution. Is there a false or untrue underlying distribution?

Authors: No, there is only "underlying distribution" for which the differential spectrum is an approximant. We remove the confusing "true" wording from the manuscript, and thank the reviewer for bringing this up.

Reviewer: 6) l. 121 Why Gaussian and why not something else? I think any distribution could be assumed. If I assumed subpopulations to exist, perhaps a Gaussian is not the best when the mean is centered on a relatively high temperature. There may be chance of sampling freezing temperatures > 0C?

Of course these can be simply removed, but this would imply a bias in the subpopulation freezing behavior.

Authors: While other distributions could be considered, the Gaussian distribution is a commonly used and well-understood distribution in statistics, making it a good starting point for modeling. Based on our fitting of experimental data, we have not observed any cases in which a Gaussian distribution resulted in the sampling of freezing temperatures above 0°C. We have now also included in the HUB code the possibility of using log-normal or left-tailed Gumbel distributions for the populations, lines 160-162:

"We also provide in the HUB code the option for the user to use the log-normal distribution, which has a tail towards higher temperatures, or the left-tailed Gumbel distribution, which has a tail towards lower temperatures."

However, in all the cases we tested, we find that Gaussians subpopulations results in the smallest error in the prediction of the cumulative spectra (this is shown in the response to reviewer 1 and the new Supp. Section S5).

Reviewer: 7) Too often in a section, the authors refer to later sections. Please minimize these instances, as it is distracting.

Authors: The structure and flow of our paper require us to reference later sections. These references serve as a guide for readers to understand the context and build up to the conclusions of the paper. We believe that minimizing these instances could potentially disrupt the coherence and clarity of our argument.

Reviewer: 8) l. 299-301 This is circular reasoning. The authors will test the droplets and IN concentrations, to test the sensitivity of Nm to the droplets and IN concentrations?

Authors: Our objective is to investigate the sensitivity of $N_m$ to the number of droplets and IN concentrations by testing the droplets and IN concentrations, measuring the freezing data, and then using the extreme value approach to compute the fraction of frozen droplets. The cumulative freezing spectra are then determined based on different combinations of number of droplets and dilutions. We then use the HUB-backward code to compare the differential spectrum of the generated Nm with the underlying distribution. We believe that this approach providse valuable insights into the relationship between Nm and droplets/IN concentrations. There is no circular reasoning.

Reviewer: 9) l. 313 What is the authors definition of an "absolute calibration". How does this differ from a "calibration".

Authors: Our definition of "absolute calibration" refers to the process of determining the exact number of active ice nucleators present in the parent sample. In our study, we use the fraction of frozen droplets that freeze homogeneously to estimate this value. Based on Poisson statistics, we can infer that there is approximately 1 active ice nucleator per droplet when this fraction is 0.6 (Figure 2A), as determined by the equation $(1-e^{(-x)})=0.6$, where x is approximately 1. With this information and the dilution factor from the parent sample, it is possible to determine the absolute concentration of IN in the parent. This is different from a general "calibration," which may refer to any process of setting or adjusting measurement parameters. We hope this clarifies the terminology used in our manuscript.

Reviewer: 10) l. 374-375 What is important about looking at a log or linear scale for the y-axis of a graph. If a graph looks better or worse on either scale, what is this telling the reader? This should be clarified.

Authors: Both representations have (of course) the same information. We had indicated in the manuscript the purpose of this dual representation in this example, lines 450-452:

"Note that we use a logarithmic scale to represent this $n_m^{optimized}(T)$ because the population corresponding to class A accounts for less than 0.1% of the total (**Table 2**)."

Reviewer: 11) l. 377-379 What is the authors quantitative criteria for "almost identical" and "unnecessary"? How much data variability is explained when two, three or more subpopulations are included? Is the number of subpopulations sufficient when 99% of the variability is explained?

On the other hand, could two different types of ice nucleating particles exist (different populations) in the same drop, but have the same distribution? This code then would mistake these 2 subpopulation as a single subpopulation. This would then misrepresent the ice nucleating subpopulations?

Authors: We are not using a quantitative criterion here for "almost identical", only that the differential spectra retrieve does not show new peaks or a shift in the existing peaks. The users of the HUB-backward code can decide from the results and the purpose of the optimization whether it is worth adding more populations (if they do not decrease the mean square error, there is no doubt that the addition of new populations is unnecessary). Our purpose in this manuscript is not to address the details of the population spectra in the examples selected but to illustrate how the method can be used for the analysis.

The HUB code does not identify the microscopic origin of the populations, only their freezing signatures. If two distinct (chemical, biological) populations of IN have the same distribution of freezing temperatures, they are considered a single population from the point of view of the analysis. However, they may be separated by changing the cooling rate (as seen in the increasing separation of the first and second peak of the cholesterol sample with decreasing cooling rate in Figure 10b), or by processing of the data (e.g. filtration, chemical treatment, etc).

Reviewer: 12) l. 424-425 Here, is it assumed that pH can change the position, width and amplitude of the distributions. This is certainly important, but I am wondering how valid is the assumption that pH only changes the amplitude, but keeping the mean and standard deviation the same? As the authors prepare their resubmission and include an uncertainty analysis, I would highly recommend the authors to fit the ice nucleation data for all pH for a common mean and standard deviation, allowing only the amplitude to be a function of pH. Then evaluate if the result is somehow within the predicted and experimental error. One could surmise that a surfaces ability to nucleate ice may or may not be pH dependent, but perhaps pH would destroy active sites instead.

Authors: Thank you for your feedback and suggestions. We agree that it is important to rigorously test the assumption that pH only changes the amplitude, while keeping the mean and standard deviation the same. Below wee show the results of fitting the ice nucleation data for all pH values using a common mean and standard deviation, allowing only the amplitude to be a function of pH. We find a poorer agreement between this set of fits and the experimental data than the one used in the manuscript.

Keeping $T_{mode,1}$ = -9.3, $s_1$ = 0.93, $T_{mode,2}$ = -4.1, $s_2$ = 0.43 for the three ice nucleation spectra, we obtain $c_2$ = 1.31 x $10^{-3}$ for pH = 6.2, $c_2$ = 1.01 x $10^{-5}$ for pH = 5.6 and $c_2$ = 1 x $10^{-10}$ for pH=4.4.

[Figure]

We now comment on this result on the manuscript and also clarify that the experimental data indicates that the pH does not change the number of IN sites, lines 474-484:

"To further illustrate the use of HUB-backward, **Fig. 10** shows the effect of pH on the subpopulations in the modes, spread and weighs that contribute to the nucleation spectrum of *P. syringae* (Snomax®), using data from (Lukas et al., 2020). Freezing in the temperature range of class A drops about 3 orders of magnitude when the pH is lowered from 6.2 to 4.4, (**Fig. 10B**). However, we note cumulative number of IN is preserved in the experimental data that the cumulative freezing spectrum (Lukas et al., 2020), indicating that the change in pH did not impact the number of nucleants. **Fig. 10C-D** demonstrates that the distributions associated with both subpopulations shift to lower temperatures when the pH decreases, and the range of freezing temperatures in class A becomes broader. An attempt to fit the cumulative spectra of Snomax at different pH with the same subpopulations, allowing only for adjustment of their weights, resulted in a poor fit to the experimental $N_m(T)$, supporting the conclusions of (Lukas et al., 2020) of a central role of electrostatic interactions in the assembly of the bacterial ice nucleating proteins and their ability to bind to ice. This analysis exemplifies how HUB-backward can be applied to quantify the dependence of IN on environmental variables."

Reviewer: 13) Please check references for consistency with doi format, URLs, use of italics, use of the correct journal and journal abbreviations.

Authors: Thank you for your feedback. We have reviewed our references and made the necessary updates to ensure consistency with the doi format, URLs, use of italics, and correct journal and journal abbreviations.

Reviewer:  (list of references provided by reviewer 2)

Fahy, W. D., Shalizi, C. R., and Sullivan, R. C.: A universally applicable method of calculating confidence bands for ice nucleation spectra derived from droplet freezing experiments, Atmos. Meas. Tech., 15, 6819-6836, 10.5194/amt-15-6819-2022, 2022a.

Fahy, W. D., Maters, E. C., Giese Miranda, R., Adams, M. P., Jahn, L. G., Sullivan, R. C., and Murray, B. J.: Volcanic ash ice nucleation activity is variably reduced by aging in water and sulfuric acid: the effects of leaching, dissolution, and precipitation, Environ. Sci.: Atmos., 2, 85-99, 10.1039/D1EA00071C, 2022b.

Herbert, R. J., Murray, B. J., Whale, T. F., Dobbie, S. J., and Atkinson, J. D.: Representing time-dependent freezing behaviour in immersion mode ice nucleation, Atmos. Chem. Phys., 14, 8501-8520, 10.5194/acp-14-8501-2014, 2014.

Knopf, D. A. and Alpert, P. A.: Water Activity Based Model of Heterogeneous Ice Nucleation Kinetics for Freezing of Water and Aqueous Solution Droplets, Faraday Discuss., 165, 513-534, 10.1039/C3FD00035D, 2013.

Vali, G.: Quantitative Evaluation of Experimental Results an the Heterogeneous Freezing Nucleation of Supercooled Liquids, J. Atmos. Sci., 28, 402-409, 10.1175/1520-0469(1971)028%3C0402:QEOERA%3E2.0.CO;2, 1971.

Vali, G.: Revisiting the differential freezing nucleus spectra derived from drop-freezing experiments: methods of calculation, applications, and confidence limits, Atmos. Meas. Tech., 12, 1219-1231, 10.5194/amt-12-1219-2019, 2019.

Wright, T. P. and Petters, M. D.: The role of time in heterogeneous freezing nucleation, J. Geophys. Res.-Atmos., 118, 3731-3743, 10.1002/jgrd.50365, 2013.

Yankofsky, S. A., Levin, Z., Bertold, T., and Sandlerman, N.: Some Basic Characteristics of Bacterial Freezing Nuclei, Journal of Applied Meteorology and Climatology, 20, 1013-1019, 10.1175/1520-0450(1981)020<1013:Sbcobf>2.0.Co;2, 1981.

---

## Author Response (AR2)

**Referee #2**:

We thank the reviewer for their comments and suggestions, which we have used to improve the manuscript. Below, we respond to each comment and describe the modifications implemented in the revision (highlighted in red in this response and the annotated manuscript for review).

Reviewer: The revised manuscript of Ingrid de Almeida Ribeiro et al. shows improvement from their previous submission. I am overall happy with their response in most aspects, but find they did not go far enough with text changes to major comments. I would consider acceptance only after following the comments are addressed.

Authors: We thank the reviewer for taking the time to review our revised manuscript. We are pleased to hear that they have noticed an improvement from our previous submission and appreciate their feedback on the text changes and comments in our manuscript.

Reviewer: i) In terms of theory, I think it is necessary to state the author's comment found in their responses, "We want to make clear here that the theoretical interpretation of these differential spectra (and their fully sampled underlying distribution of nucleation temperatures) is not a goal of the present study." Often the authors claim a connection between their HUB models and theory/physical analysis/kinetic models/and nucleation mechanisms. Without this statement, there is too much confusion for a reader. The author's comment should be stated in the manuscript.

Authors: The paper has two clearly stated and explicit goals in the introduction:

"The first goal of the present study is to provide a strategy to optimize the sampling of drop-freezing experiments to derive interpretable differential spectra that is a good approximant of the underlying distribution of heterogeneous ice nucleation temperatures of the sample."

"The second aim of our study is to map the cumulative freezing spectrum $N_m(T)$ into the differential spectrum $n_m(T)$, in terms of subpopulations that may correspond to different physical nucleation sites in the sample."

Thus we interpret that the sentence the reviewer fears may confuse the readers is the one that indicates that the analysis performed with the methodology presented here can be interpreted with theory, models, etc. We now replace "further enables" by "could further enable" in the sentence that the reviewer found confusing and also refer the readers to two recent studies in which we do indeed use the results of HUB-backward together with classical nucleation theory for finite surfaces to predict the size of the IN in lichen and fungi:

"The determination of distributions obtained from the HUB-backward code could further enable the interpretation of the experimental ice nucleation spectra with size and structure of INs using nucleation theory, kinetic models, and molecular simulations. For example, (Schwidetzky et al., 2023) illustrates the use of the distribution of freezing temperatures obtained with HUB-backward together with classical nucleation theory for finite surfaces to interpret the size of the IN of *Fusarium acuminatum*."

Reviewer: ii) The sentence in their abstract, l. 21-23, "The differential spectrum computed with HUBbackward is an analytical function that can be used to reveal and characterize the underlying number of IN subpopulations of complex biological samples", is misleading. It is claimed by the authors in their response, it is not possible to know the subpopulations. Therefore, the HUB does not reveal and characterize the underlying number of IN subpopulations, the authors must prescribe them. I still find the authors overstate what the HUB does and does not do. I would like the authors to claim in the revised

manuscript, as they claim in their response and remove any mention that HUB reveals/characterizes subpopulations.

Authors: We stand by the claim that ""The differential spectrum computed with HUB-backward is an analytical function that can be used to reveal and characterize the underlying number of IN subpopulations of complex biological samples". It is true that the user has to set the maximum number of subpopulations to be adjusted, but if the number is insufficient, the user will note the disagreement between the calculated and measured cumulative spectrum. If, instead, the proposed number of populations of freezing temperatures is too high, the user will notice that they are overlapped and not truly distinguishable.

Reviewer: iii) The added sentence, "However, the requirement of a model for the freezing rates and their distribution across the sample hinder their interpretability and accuracy at reproducing the experimental freezing curves, particularly in complex samples containing multiple populations." I think would be highly contended by the Ben Murray group cited in the revised manuscript. They have used multi-component freezing rates across a sample and have accurately reproduced experimental freezing curves. I would remove this sentence on l. 59-61.

Authors: We appreciate the reviewer's concern and remove the issue of accuracy from the the sentence and rewrite it as "These approaches require an analytical model for the freezing rates and their distribution across the sample."

Reviewer: iv) In Sear (2013), Fig. 1A, a cumulative spectra/frozen fraction based on extreme value statistics is shown in an ice nucleation model. The authors of the manuscript in question make two claims in their response. 1) "note that –to our knowledge extreme-value statistics has not been previously used for the modeling of ice nucleation data, lines 77-93:" and 2) "To our knowledge, the fraction of frozen droplets and the cumulative spectra based on extreme value sampling have not been explored in any previous study." I can agree the manuscript provides a unique perspective, but the claims 1) and 2) made by the manuscript authors are not correct. The title of Sear (2013) even states "…a model for ice nucleation". Please remove these statements and similar and provide a better in-text description of the Sear (2013) citation and what was done previously for modeling ice nucleation using extreme value statistics.

Authors: Figure 1 A of Sear shows cumulative distribution functions obtained from the three generalized extreme value (GEV) distributions, i.e. Gumbel, Frechet and Weibull. These theoretical cumulative fractions obtained from one of these GEV correspond to the fraction of frozen droplets only at certain conditions that are not generally satisfied in experiments:

- The sampling of the droplets has to be complete. However, experiments typically sample tens to hundreds of droplets (and we show that depending on the sampling per dilution the same number of droplets may produce different results).

- The number of IN the droplets has to be extremely large and constant. However, in experiments there are fluctuations (that we model with the Poisson distribution) and –depending on the dilution- the number of IN per droplet spans all the range from reasonably concentrated to a few. Figure 2C shows that if the number of IN per droplet is not extremely large, extreme value sampling produces functions that are intermediate between the underlying distribution and the limit at infinite size; thus a GEV is only the limit for extremely concentrated solutions and we show in this study that sampling the more dilute concentrations is key to reconstruct the underlying distribution.

- As Sear himself notices in his paper, there is no analytical theory to predict the GEV of samples that combine distributions of nucleation temperatures (e.g. those of *Ps. syringae*, pollen, cholesterol, etc).

Our study builds on the same fundamental assumptions of Levine and Sear, but takes a numerical approach to the extreme value sampling that makes it suitable for a finite number of droplets, the full range of dilutions, and to model the outcome from underlying distributions of IN temperatures with multiple peaks. We now elaborate more on the work by Levine and Sear and what are the challenges they cannot address, which are the motivation for the approach in our work (lines 85-103):

"A central assumption of the singular freezing approximation is that the freezing of a droplet containing multiple INs is promoted by the IN with the highest nucleation temperature (Levine, 1950). The extreme-value sampling is apparent in the concentration dependence of $f_{ice}(T)$ in experiments (Marcolli et al., 2007; Budke and Koop, 2015; Kunert et al., 2018; Lukas et al., 2022). Using probability theory, Levine demonstrated that if the distribution of ice nucleation temperatures of the IN population follows an exponential distribution, then the sampling of droplet freezing temperatures corresponds to a Gumbel distribution, and the median freezing temperature $T_{MED}$ of the droplets scales with the logarithm of the number (or total nucleating area) of IN per droplet (Levine, 1950). Sear more recently demonstrated that Levine's approach is a particular solution for a generalized extreme-value problem, and used modern extreme value statistics to derive the scaling of $T_{MED}$ with the number of IN sites per droplet for the three generalized extreme value distributions (GEV): Gumbel that would arise from an underlying IN distributions with exponential tails, Frechet from those with power law tails, and Weibull from those with an upper cutoff in the freezing temperature of the IN (Sear, 2013). However, there are limitations for the use of the analytical approaches of Sear and Levine for the interpretation of actual drop freezing data. First, the extreme value sampling results in one of the three GEV only in the limit of extremely large number of IN per droplet, while in experiments the sampling is typically performed over dilutions down to a few IN per droplet. There is no analytical formulation for the dependence of the extreme value distribution in the low to intermediate concentration regime. Second, the analytical theory assumes that the sampling is complete (i.e. the number of droplets is extremely large), while experiments are typically performed with tens to hundreds of droplets. Third, Sear notes that there is no general analytical theory to predict the GEV from a mixture of populations of nuclei with different temperature dependences (Sear, 2013). In this study we overcome these three limitations through a numerical implementation of extreme-value statistics for the modeling of drop-freezing experiments."

.

Reviewer: v) "Underlying": the authors must define the word, underlying, in their manuscript. There is too much risk of misconception that there are, in fact, different real populations. Their population number is assumed and distribution parameters are fitted in the manuscript. This is not "real".

Authors: We understand the concern of the reviewer, and now clarify that these subpopulations are of freezing temperatures and do not necessitate a mapping to distinct physical entities (line 151-157):

"For generality, we represent $P_u(T)$ as a linear combination of normalized continuous distributions $P_i(T)$ that represent subpopulations of freezing temperatures:
$$P_u(T) = c_1 P_1(T) + c_2 P_2(T) + \ldots + c_p P_p(T),  \qquad (2)$$
where $p$ is the total number of subpopulations, $P_1(T), P_2(T), \ldots, P_p(T)$ are normalized distribution functions, and $c_1, c_2, \ldots, c_p$ are their weights such that $\sum_{i=1}^{p} c_i = 1$. These subpopulations could correspond to different chemical, topographical or structural motifs in the IN samples, although chemically distinct species could also produce overlapped freezing signatures, and a single species could display a broad freezing range. Our formalism does not require a mapping of subpopulations of freezing temperatures to physical IN sites."